

Comparing the performances of WRF QPF and PERSIANN-
CCS QPEs in karst flood simulations and forecasting with a new
Karst-Liuxihe model
Ji Li[1], Daoxian Yuan[1,2], Aihua Hong[3], Yongjun Jiang[1], Jiao Liu[4], Yangbo Chen[5],
[1]School of Geographical Sciences of Southwest University, Chongqing Key
Laboratory of Karst Environment, Chongqing 400715, China
[2]Karst Dynamic Laboratory, Ministry of Land and Resources, Guilin 541004, China
[3]The Laboratory of Chongqing groundwater resourse utilization and environmental pr
otection (Nanjiang Hydrogeological Team Under the Chongqing Geological Bureau
of Geology and Minerals Exploration), Chongqing 401121, China
[4]Chongqing Hydrology and Water Resources Bureau, Chongqing 401120, China
[5]Department of Water Resources and Environment, Sun Yat-sen University,
Guangzhou 510275, China
Correspondence: Ji Li (445776649@qq.com)
**Abstract**
Long-term, available rainfall data are very important for karst flood simulations and
forecasting. However, in karst areas, there is often a lack of effective precipitation available to
build distributed hydrological models. Forecasting karst floods is highly challenging.
Quantitative precipitation forecasts (QPF) and estimates (QPEs) could provide rational
methods to acquire the available precipitation results for karst areas. Furthermore, coupling a
physically-based hydrological model with the QPF and QPEs felicitously could largely
enhance the performance and extend the lead time of floods forecasting in karst areas, the
performance of coupling the Weather Research and Forecasting Quantitative Precipitation
Forecast (WRF QPF) and Precipitation Estimations through Remotely Sensed Information
based on the Artificial Neural Network-Cloud Classification System (PERSIANN-CCS
QPEs) with a new fully distributed and physical hydrological model, the Karst-Liuxihe model
in flood simulations and forecasting in karst area. This study served 2 main purposes: one
purpose is to compare the performances of WRF QPF and PERSIANN-CCS QPEs for rainfall


forecasting in karst river basins. The other purpose is to test the effective feasibility and
application of the karst flood simulation and forecasting by coupling the 2 weather models
with a new Karst -Liuxihe model. The new Karst-Liuxihe model improved the structure of the
model by adding the karst mechanism based on the Liuxihe model as follows: 1. Refine the
model structure and put forward the concept of karst hydrological response units (KHRUs) in
the model. The KHRU, as the smallest unit of the Karst-Liuxihe model, is defined in this
paper to be suitable for karst basins; 2. Increase the calculations of water movement rules in
the epikarst zone and underground river, such as the division of slow flow and rapid flow in
the epikarst zone and the exchange of water flow between the karst fissures and conduit
systems; thus, the convergence of the underground runoff calculation method is improved to
be suitable for karst water-bearing media; and 3. Add some necessary hydrogeological
parameters in the coupled model to reflect the true conditions of rainfall-runoff in the karst
underlying surface. Moreover, the flood detention and peak clipping effects due to the
upstream karst depressions during flooding were considered and reasonably calculated in the
coupled model. The flood detention effect can affect the peak flow time error simulated in the
model and make the true peak flow appear later; the flood peak clipping effect can affect the
flood peak flow relative errors and the simulation errors of floods volume. The consideration
of these 2 factors in the model makes the flood simulations and forecasting effects more
credible. The rainfall forecasting result show that the precipitation distribution of the 2
weather models was very similar compared with the observed rainfall result. However, the
precipitation amounts forecasted by WRF QPF were larger than that measured by the rain
gauges, while the quantities were smaller by the PERSIANN-CCS QPEs. A postprocessing
algorithm was adopted in this paper to correct the rainfall results by the 2 weather models.
The karst flood simulation and forecasting results showed that the flood peak flow
simulations were better by coupling the Karst-Liuxihe model with the PERSIANN-CCS
QPEs, and coupling the Karst-Liuxihe model with WRF QPF could extend the lead time of
flood forecasting largely, as a maximum lead time of 96 hours can provide an adequate
amount of time for flood warnings and emergency responses. The satisfying and rational karst
flood simulation evaluation indices proved that coupling the 2 weather models with the new
Karst-Liuxihe model could be effectively used for karst river basins, which provides great
practical application prospects for karst flood simulations and forecasting. In addition, the
postprocessing method used to revise the 2 weather models in this paper is feasible and
effective, and this method can largely improve the coupled model application effectiveness
and prospect in karst river basins.
**1 Introduction**

In karst areas, the general lack of long-term meteorological data, especially precipitation

data, is a great challenge to the simulation and forecasting of flood events based on



hydrological models (Li et al., 2019). Quantitative Precipitation Forecasts and Estimates
(QPF and QPEs) are methods that may enable precipitation data in karst river basins to be
easily obtained. The Weather Research and Forecasting (WRF) model, a type of QPF
technology, is regarded as a new generation mesoscale weather forecasting
model that could provide rainfall data with high accuracy at 1-10 km horizontal resolution
(Skamarock et al., 2005). Furthermore, the WRF QPF can forecast rainfall data with a long
lead time in karst areas, which is very important for flood warnings and mitigation because
more time is provided for flood emergency responses (Tingsanchali, 2012). In this study, the
maximum lead time is 96 hours, which can be the greatest factor of concern for decision
makers in flood forecasting (Han et al., 2007). The PERSIANN-CCS is a QPE technology by
weather satellites, which could estimate long-term and high-resolution rainfall data (Yang et
al., 2004, 2007). However, only a few studies of rainfall forecasting based on WRF QPF and
PERSIANN-CCS QPEs have been conducted in karst areas until now, and even if there are
studies, the practical accuracy is generally poor. In addition, the flood simulation and
forecasting results of coupling these weather models with hydrological models have poor
precision in karst river basins due to the system error stack of the models as well as the
complex hydrogeological conditions of karst water-bearing media (Ford and Williams, 1989;
Kovacs and Perrochet, 2011).
Generally, there are only a few rain gauges in karst river basins. Especially in the
upstream areas of the basins, which comprise mountains and valleys with complex
topographies, it is difficult to set up rain gauges to effectively obtain rainfall data. The study
area in this paper is the Liujiang basin with $5.8 \times 10^4$ km$^2$ drainage area; however, there are
only 66 rain gauges. On average, there is only approximately 1 rain gauge per 1,000 km$^2$, and
the representativeness is too weak to reflect the actual rainfall that occurs in the basin. Under
these circumstances, effective precipitation results could potentially be acquired by using
numerical weather models in karst river basins. In recent years, numerical weather prediction
models have become increasingly mature with the great progress of the 3S (the remote
sensing/RS, geography information system/GIS, and global positioning system/GPS)
technologies and can provide a global range of rainfall forecasting products with reasonable
and high precision.
The current mainstream numerical weather models include the European Centre Weather
Forecasts model (Molteni et al., 1996), the Japan Meteorological Agency weather model
(Takenaka et al., 2011), the QPEs by weather radars (Rafieei et al., 2014; Delrieu et al., 2014;
Faure et al., 2015), WRF QPF (Skamarock et al., 2008), satellite QPEs (Bartsotas et al., 2017;
Wardhana et al., 2017), and others. Among these weather models, WRF QPF and
PERSIANN-CCS QPEs may be better ways to acquire precipitation results effectively in
karst basins. The lead time of the QPF by the latest WRF model is 1-15 days (Ahlgrimm et al.,
2016). Therefore, coupling the hydrological model with WRF QPF for floods warning and



forecasting, the lead time could be extended greatly (Zappa et al., 2010). In comparison to this
model, the observed precipitation by rain gauges has no lead time because the precipitation
has already fallen to the ground. The lead time of WRF QPF in this study was 96 hours. That
is, the equivalent of a 96-hour lead time of flood forecasting, which is very important for the
safe transfer of people and property before the floods. PERSIANN-CCS QPEs could offer
reasonable rainfall data with high precision, and coupling this model with the distributed
hydrological model gave good results in karst flood simulations (Ji et al., 2019).

Several scholars at home and abroad have achieved acceptable results using numerical

weather models (Hu et al., 2013; Stenz, 2014; Bartsotas et al., 2017; Wardhana et al., 2017).
However, some uncertainty remains that cannot be neglected in the model application, which
results in the poor precision of these weather models (Goudenhoofdt and Delobbe, 2009). In
this study, 2 effective measures could be used to reduce the uncertainty and improve the
precision of the weather models in the karst river basins. One is to choose a suitable model
spatial resolution, which could largely affect modelling effects. A initial spatial resolution for
WRF QPF and PERSIANN-CCS QPEs are 20 km$\times$20 km and 0.04 °$\times$0.04 °, respectively.
After many tests, the best spatial resolution for the 2 weather models in the study area is 200
m$\times$200 m, which can well match the hydrological model in this paper. The other measure is to
reduce the systematic errors of the weather models. A postprocessing algorithm was proposed
in this paper to correct WRF QPF and PERSIANN-CCS QPE results in the karst area, which
could reduce the rainfall result uncertainties and make the results easier to receive and more
credible.

A hydrological model, as a physics-mathematics computational tool, is an important

method used to accurately simulate and forecast flood events. Where the precipitation occurs,
which is the hydrological model input data, could be the driving factor in flood forecasting
(Li et al., 2017). Coupling a hydrological model with WRF QPF and PERSIANN-CCS QPEs
has a great capacity and prospect for floods simulations and forecasting in karst areas.
However, the traditional hydrological models such as lumped models have considerable
disadvantages in karst flood simulations and forecasting. The complex hydrogeological
conditions and highly anisotropic karst aquifers as well as water-bearing media in karst areas
cause flood processes to be more complex and nonlinear than those in non-karst basins
(Goldscheider and Drew, 2007; Hartmann et al., 2013). Lumped hydrological models have a
simple model structure, and only a few hydrogeological data are required for modelling.
These models usually treat the catchment as a whole unit and ignore the spatial variations in
rainfall-runoff as well as the complexity of the underground space structure of karst aquifers
(White, 2007). Additionally, the lumped model parameters are homogenized or generalized,
and the same set of parameters are adopted for the whole basin, which results in poor
precision of flood forecasting applications in karst areas (Scanlon et al., 2003). Physically
based distributed hydrological models have great application potential and capabilities in


improving the performance of karst flood event forecasting than lumped hydrological models
(Ambroise et al., 1996). In a karst river basin, the entire basin could be divided into many grid
units known as the karst sub-basins by the DEM data in the distributed models, and by
coupling the grid rainfall with WRF QPF and PERSIANN-CCS QPEs, the actual karst
development characteristics and rainfall-runoff processes can be precisely reflected.
Therefore, the distributed hydrological models are better than the lumped models for flood
simulations and forecasting in karst river basins. To improve the performance and precision,
in this study, the karst subbasins will be further divided into smaller grid units known as karst
hydrology response units (KHRUs) in the distributed hydrological model.
Shustert and White (1971) made a good attempt to use a distributed model in karst areas.
After that, an increasing number of distributed models have been used in karst flood
forecasting (Quinlan and Ewers, 1985; Ambroise et al., 1996; White, 2002, 2005, 2007;
Gallegos et al., 2013). Ghasemizadeh (2012) introduced several commonly used distributed
hydrological models and their application effects in karst watersheds. However, there are 2
obvious shortcomings with the distributed hydrological models when used in karst areas. One
is the problem of an insufficient data supply. In particular, it is highly challenging to build
distributed models because of the lack of necessary hydrogeological data. The other is the
problem of model calculation efficiency. In general, there are many parameters in the
distributed models, which require many computational resources, which leads to low
efficiency (Chen et al., 2017). In this paper, the hydrogeological data problem is solved by a
field survey and tracing test as well as a drill-hole pumping test. In addition, the property data
of the study area, including the DEM data, the soil types and the land use types, could be
downloaded expediently from the internet at no cost. An improved Particle Swarm
Optimization method (Chen et al., 2016) was used for parameter optimization, and the use of
this algorithm could improve the computing efficiency of the distributed model and reduce
the uncertainty in the parameters.
Currently, there is no unified, widely agreed upon and highly practical distributed karst
hydrological model being used around the world. Some distributed models may work
accurately in the local area but may not be transferable to another karst basin. Moreover, no
such model with high precision could be generally applicable to a typical karst watershed in
southwest China, where karst is the most developed. Therefore, we hope to find a distributed
hydrological model that has general applicability to the karst area in southwest China through
the application of the model proposed in this study. In this paper, the feasibility and
application effects of coupling a new karst hydrological model, i.e., the Karst-Liuxihe model
with WRF QPF and PERSIANN-CCS QPEs in karst floods simulations and forecasting are
studied. Conducting this study served 2 purposes: one purpose was to synthetically compare
the performances of WRF QPF and PERSIANN-CCS QPEs in rainfall forecasting in the
study area. The other purpose was to verify the performance and feasibility of karst flood



simulations by coupling the 2 weather models with the new Karst-Liuxihe model. The new
Karst-Liuxihe model is improved by adding the karst mechanism based on the Liuxihe model
prototype (Chen, 2009). The improvements are described below: (1) The karst water-bearing
medium is simplified in the model settings. (2) The model structure is refined, as the minimal
model structure is divided into KHRUs in this study. (3) The karst mechanism is added to the
model calculation, where the calculation principle of the fluid migration rule in the epikarst
zone is increased, including the flow movement rule in the shallow karst fissure network; the
unsaturated zone, the rapid flow and the slow flow in the model are divided, and the hydraulic
relationship between the karst fissure and the conduit systems is calculated. (4) The
calculation principle of the groundwater confluence to the basin outlet is improved. (5) Some
necessary hydrogeological parameters that are suitable for karst aquifers are added to the
model, including the permeability coefficient $K$ and so on. There are 14 parameters in the
original Liuxihe model, and the parameter number increased to 20 in the Karst-Liuxihe
model.
In this study, both weather models, i.e., WRF QPF and PERSIANN-CCS QPEs, can
provide high-resolution grid rainfall data, which are coupled with the Karst-Liuxihe model
could make a satisfactory effect in karst floods simulations and forecasting. This model is
applied to the Liujiang karst basin, which is the area of China where karst is the most
developed. The karst flood simulation effect of the coupled model is excellent. In particular,
the simulation error of the flood peak flow is effectively controlled. Moreover, the maximum
lead time of rainfall forecasting can reach 96 hours, which makes a significant difference for
flood warnings and the secure transfer of people and property before the occurrence of
flooding. The coupling proposed in this study could be applied to other karst river basins in
China and even around the world due to the reasonable and acceptable flood simulation
effects.
**2 Study area and data**
2.1 Geology and landforms

The study area of this paper is the Liujiang karst river basin, which located at
23.9 °~24.5 °N, 108.9 °~109.7 °E in southwest China. The channel length of Liujiang river is
about 1,120 km and the area is about $5.8 \times 10^4$ km². It is the most developed karst basin of
China, as shown in Fig. 1a, the map of Liujiang watershed. The carbonate rocks distribution
area is about $1.9 \times 10^4$ km², which are mainly distributed in the northern part of the watershed.
The peak forest plain in the downstream basin and the peak cluster depression in the middle
and upper reaches are the dominant landforms of the study area. The karst valley is the main
landform in the south, where the underlying bedrock, which mainly comprises carbonate and
dolomite. A large area of limestone is distributed in the western part, where the peak cluster
depression is dominant. Hilly and mountain are the dominant landforms in the eastern part. In



particular, the highest mountain in the basin is Leigong Mountain, which has an elevation of
2124 m (as shown in Fig. 1b) and is located in the northeast basin. The dominant landforms in
the central part and downstream are the peak forest plains.
Figure 1. The sketch map of Liujiang karst watershed.
The upstream area of the basin is located in the southern part of the ancient
Paleocaledonian fold belt and the southeastern edge of the southwest China depositional area,
where a large area of sedimentary rock is distributed. The outcrop strata in the basin are
ancient and intact and mainly include Sinian, Cambrian, Silurian, Ordovician, Upper
Devonian, Lower Carboniferous, Upper Permian, Lower Triassic, Paleogene, Quaternary
Pleistocene and Holocene.
After a long karst landform evolutionary process, karst development in the basin is now
very mature. At first, there were mainly small karst doline funnels in the basin; then, the
landform evolved into a peak cluster depression (as shown in Fig. 2, photographs of the
middle and upper reaches) as carbonate rocks continued to be eroded by karst water as well as
the fluviraption of allogeneic water, especially the Liujiang River. Under these interior
erosional effects and exterior fluviraption for so many years, the geomorphological evolution
reached an old age, i.e., the peak cluster depressions had evolved to the peak forests
(Williams, 1987), especially in the downstream (as shown in Fig. 2, photograph of the
downstream reaches).
Figure 2. The karst landform evolution of the Liujiang basin.
2.2 Precipitation, karst flood and property data
The Liujiang River, a rain-source river, the average annual precipitation in the basin is
between 1400 and 1700 mm. The flood season is from May to September, and the flood
volume can account for 80% of total runoff. The maximum peak flow is $2.59 \times 10^4$ m$^3$ s$^{-1}$ (in
2009, as shown in Fig.12 in the section 6.3). The water level rise over a 24-hour period can be
as high as 12.1 m (in 1978). The mean annual maximum flood peak discharge is 15,200 m$^3$ s$^{-1}$,
and the maximum 7-day mean flood volume is 5.38 billion m$^3$. In the upper reaches, most of
the landforms are deep-cut canyons shaped like a "V" except in the river source regions. The
elevation of these canyons is usually greater than 1000 m with a relative height of 500~700 m
(as shown in Fig. 1b). In these canyons, the runoff responds quickly to rainfall, and the area is
prone to regional flood disasters.
The flood characteristics are closely related to rainstorms, the watershed topography and
the karst landform. Larger floods are mostly multipeak processes, and an increase lasts only a
short period of time, i.e., the flood peak occurs quickly and recedes quickly in terms of the
flood response, which usually causes considerable damage. In the 1990s, the frequency and





intensity of rainstorms and flood disasters were increasing with the increase in extreme
weather. The north-eastern and western areas of the basin are the main flood sources, and this
the area where the most developed karst is located. Especially the karst conduits are well
developed in the underground aquifer. According to the tracing test conducted in Liujiang
basin, during the flood season, the flood velocity can reach to 43-130 km/d. The maximal
velocity is 173 km/d, which indicates the karst underground rivers are well developed in the
study area. The karst features can significantly affect the hydrologic process, especially
during the rainfall-runoff process in the model. It is highly challenging to accurately simulate
the karst water cycle rules and forecast the floods changeable trends in the future.
In the study area, there are total of 66 rain gauges, 156 grid gauges for WRF QPF and 131
grid gauges for PERSIANN-CCS QPEs (as shown in Fig. 1a), respectively. And 5 floods that
occurred from 2008-2013 were used to verify the performance of coupling the Karst-Liuxihe
model with WRF QPF and PERSIANN-CCS QPEs. Hourly precipitation from the rain gauges
was adopted to revise the products of the 2 weather models in this paper. The property data of
the watershed are mainly the DEM data, the soil types as well as the land use types. These
property data could be downloaded easily from the internet at no cost: (1) The DEM data are
from http://srtm.csi.cgiar.org, last accessed: 02 April 2019. (2) The land use types can be
downloaded from http://landcover.usgs.gov, last accessed: 02 April 2019. (3) The soil types
are from http://www.isric.org, last accessed: 05 April 2019. After resampling in the ArcGIS
10.2, these property data are downscaled to the same resolution as the hydrological model in
this paper.
**3 WRF QPF and PERSIANN-CCS QPEs**
3.1 WRF QPF12
The WRF QPF used in this study was the WRF Advanced Research model version 3.4
(Skamarock et al., 2008), which is a 3-dimensional and nonhydrostatic system that can
forecast complex weather changes on cloud scale and synoptic scale well. This model is
especially precise at 1-10 km horizontal resolution, which can satisfy the practical application
requirements of rainfall forecasting in this study. WRF QPF was applied in this study using
the following configurations: (1) The domain of the WRF QPF model is set at 24 °N and
109 °E, as the location of the basin is 23.9 °~24.5 °N, 108.9 °~109.7 °E. (2) The vertical
structure of the model includes 28 levels with the Lambert conformal projection (Li et al.,
2015). (3) The initial temporal and spatial resolutions were 3-hour and 20 km×20 km,
respectively. Following downscaling, the temporal and spatial resolutions were 1-hour and
200 m×200 m, respectively. The downscaled method, which was calculated in ArcGIS 10.2
through the statistical scales relationship between the DEM data and weather model (Fan et
al., 2017). (4) The entire basin was covered by 156 grid gauges based on the WRF QPF. The





rainfall forecasting was produced with a lead time of 96 hours (other results of lead times
such as 24, 48 and 72 h have also been calculated (Li et al., 2017)). (5) The WRF QPF results
were evaluated and revised by comparing the rainfall data from the rain gauges.
The WRF QPF parameters were set according to the following configurations: (1) The
single-moment, 3-class microphysics parameterization is used in this study (Hong and Lim,
2006). (2) The Yonsei University (YSU) planetary boundary layer scheme and the Kain-
Fritsch cumulus parameterization (Kain, 2004) are adopted to optimize the cumulus
parameters. (3) Other physics schemes for the model parameters used in this paper include the
Goddard scheme (Chou and Suarez, 1994), Rapid Radiative Transfer Model (Mlawer et al.,
1997) and the NOAH scheme (Ek et al., 2003). More details on the WRF QPF model and its
parameter settings can be found in the research results of previous studies (Li et al., 2015; Li
et al., 2017).
3.2 PERSIANN-CCS QPEs
The PERSIANN-CCS QPEs (Yang et al., 2004, 2007), which is developed based on the
PERSIANN prototype system (Hsu et al., 1999); this system is a next-generation rainfall
estimation system based on geostationary satellites that use computer imaging technology and
pattern recognition technology. The PERSIANN-CCS QPE system was based on
geostationary infrared imagery and daytime visible imagery (Soroosh et al., 2000). The
system is automated for estimating precipitation through the use of satellite remote sensing
technology. The parameters of the PERSIANN system could be optimized efficiently by a
self-adaptive artificial neural network (Yang et al., 2007).
The model setup, parameter optimization and rainfall estimation procedures of
PERSIANN-CCS (Hsu, 2007; Li et al., 2017) can be found in operating manuals and user
guides from http://chrs.web.uci.edu/projects_nasa.php, last accessed: 15 April 2019. However,
in practical application, the PERSIANN-CCS QPE model does not have to be built to obtain
the rainfall data in a particular study area. Worldwide products of QPEs based on the
PERSIANN-CCS including the rainfall results in this paper could be easily downloaded at no
cost from http://cics.umd.edu/ipwg/us_web.html, last accessed: 18 March 2019. Therefore,
the rainfall data from the PERSIANN-CCS QPEs could be obtained expediently in karst areas
where rain gauges are usually lacking.
The specific operational steps for the PERSIANN-CCS QPEs in this study area are as
follows: (1) Determine the time and scope of the study area, i.e., the rainfall occurrence and
end time as well as the location according to the longitude and latitude. (2) Download the
estimated precipitation data by the PERSIANN-CCS. (3) Analyze and appraise the products
of PERSIANN-CCS QPEs by comparing the observed rainfall by rain gauges. (4) Revise the
PERSIANN-CCS QPEs products by using appropriate methods.
The PERSIANN-CCS QPE products can generate precipitation data at a time interval of





30 min and a spatial resolution of 0.04 °×0.04 °(Yang et al., 2007). The spatial resolution was
downscaled to 200 m×200 m using a downscaling method (Fan et al., 2017) to suit the
resolution of the Karst-Liuxihe model in this paper. The time interval was changed to 1 hour.
3.3 Forecasting and evaluation of the precipitation results

There are total of 66 rain gauges, 156 grid gauges of WRF QPF and 131 grid gauges of

PERSIANN-CCS QPEs in this study area, respectively. These grid gauges can cover the
entire basin (as shown in Fig. 1a) and provide a representative rainfall product. The WRF
QPF model offers rainfall forecasting with a lead time of 96 hours, while the rainfall
estimation results of PERSIANN-CCS have no lead time. The hourly precipitation data for
2008, 2009, 2011, 2012 and 2013 from the products of the 2 weather models were produced,
compared and revised in this study by using the observed precipitation data of rain gauge.
The forecasting, estimation and comparison of the rainfall results by the 3 precipitation
products, i.e., the WRF QPF model, the PERSIANN-CCS QPEs, and the rain gauge
precipitation are shown in Figs. 3, 4, 5, 6 and 7, respectively.

Figure 3. The rainfall results of the 3 precipitation products (2008).

Figure 4. The rainfall results of the 3 precipitation products (2009).

Figure 5. The rainfall results of the 3 precipitation products (2011).

Figure 6. The rainfall results of the 3 precipitation products (2012).

Figure 7. The rainfall results of the 3 precipitation products (2013).

Figs. 3-7 showed the average value of the rainfall results of the WRF QPF model, the

PERSIANN-CCS QPEs, and the rain gauge precipitation, where (a), (b), and (c) are the
average values of the rainfall results according to the rain gauge, WRF QPF, and
PERSIANN-CCS QPEs, respectively. (d) and (e) are the quantile-quantile plot, a 45-degree
line here is drawn to compare the rainfall results of the 2 weather models and the rain gauge
precipitation, respectively.

According to the results shown in Figs. 3-7, the rainfall distributions appeared to be quite

similar with WRF QPF, the PERSIANN-CCS QPEs, and observed precipitation by rain gauge.
Especially from Figs.3-7 (d) and (e), the 2 precipitation plots, i.e., WRF QPF and the rain
gauge precipitation, PERSIANN-CCS QPEs and the precipitation by rain gauge were very
closely distributed around the 45-degree lines, meant the distribution of these 3 rainfall
products were close to one another. However, a relative error of the 3 rainfall products cannot
be ignored. The results from the WRF QPF were larger than those from the rain gauges, while
the PERSIANN-CCS QPEs were smaller, which meant that relative errors exist between the
weather model precipitation values and the rain gauge precipitation.

To further quantitatively evaluate and compare the rainfall results of the 2 weather

models with the rain gauge precipitation, the average precipitations of the 3 rainfall products
were listed in Table 1.





Table 1. The quantitative rainfall comparison results of the 3 precipitation products.
From the rainfall results listed in Table 1, some relative errors between the 2 weather
models and the rain gauge precipitation cannot be ignored. The average precipitation values
of WRF QPF were larger than the rain gauge precipitation, while the PERSIANN-CCS QPEs
values were smaller. The relative errors between the PERSIANN-CCS QPEs and the
precipitation by rain gauge were less than those of the WRF QPF and the rain gauge
precipitation. The rainfall estimation results according to PERSIANN-CCS had no lead time,
while the WRF QPF model offered rainfall forecasting with a lead time of 96 hours, which
meant a lead time of 96 hours for flood forecasting by coupling the Karst-Liuxihe model with
WRF QPF model in this study.
The average relative errors were 17% and -14% for WRF QPF and PERSIANN-CCS
QPEs, respectively. These errors are considerable relative errors and cannot be ignored.
Therefore, an effective method should be used to reduce these relative errors and make the
rainfall results by the 2 weather models more credible and receivable.
3.4 Postprocessing of the 2 weather models
To make the quantitative values of the rainfall results from WRF QPF and PERSIANN-
CCS QPEs closer to those of the observed precipitation by rain gauge, which means to make
the forecasting rainfall results are more credible, the precipitation products according to the 2
weather models were revised using the rain gauge precipitation that was considered as the
true precipitation of the basin. The procedures of postprocessing the 2 precipitation products
are as follows.
1. The average values of WRF QPF and PERSIANN-CCS QPEs were calculated according to
this equation:

$$\overline{P}_{\text{WRF/PERSIANN-CCS}} = \frac{\sum_{i=1}^{N} P_i F_i}{N} \qquad (1)$$


where $\overline{P}_{\text{WRF/PERSIANN-CCS}}$ are the average values of the precipitation results based on WRF QPF
and PERSIANN-CCS QPEs, $P_i$ is the precipitation of the 2 weather models at i grid gauge,
$F_i$ are the watershed areas of i grid gauge, and $N$ are the grid gauges numbers.
2. Average values of the observed precipitation based on rain gauge by this equation:



$$\bar{P}_2 = \frac{\sum_{j=1}^{M} P_j}{M} \qquad (2)$$

where $\bar{P}_2$ are the average values of the rain gauge precipitation, $M$ are the rain gauge numbers,
and $P_j$ are the average values of the observed precipitation of j rain gauge.
3. Average values of the rain gauge precipitations were adopted to correct the WRF QPF and
PERSIANN-CCS QPEs using this equation:
$$P_i^{'} = P_i \frac{\bar{P}_2}{\bar{P}_{\text{WRF/PERSIANN-CCS}}} \qquad (3)$$

where $P_i^{'}$ is the quantitative value of the precipitation according to WRF QPF and
PERSIANN-CCS QPEs after revision at i grid gauge, and $P_i$ are the precipitation values of
the 2 weather models at the $i$ grid gauge.
This postprocessing method made the rainfall results based on the PERSIANN-CCS
QPEs and WRF QPF closer to the observed rainfall results by rain gauges, which can largely
reduce the systematic errors of the 2 weather models. Therefore, the revision method
described in this study was feasible. After the postprocessing, the precipitation products based
on the 2 weather models were fed into the Karst-Liuxihe model to validate the model's
feasibility in karst flood events simulations and forecasting in the study area.
**4 Hydrological model**
4.1 The Liuxihe model
The Liuxihe model, a fully physically-based distributed hydrological model, was
proposed by Y, Chen (Chen, 2009), and this model earned its name through the first
significant successes in flood forecasting in the Liuxihe River basin, Guangdong Province,
China. The Liuxihe model has achieved many reasonable and gratifying research results in
the past decade (Chen, 2009, 2018; Fan et al., 2012; Liao et al., 2012; Chen et al., 2016, 2017),
which is especially significant for flood forecasting in some reservoirs and catchments (Li et
al., 2017, 2019; Hui et al., 2018).
The entire structure of Liuxihe model is divided into 7 sub-models, including the
watershed delineator and data mining sub-model, the unit classification and section
estimation sub-model, the rainfall fusion calculation sub-model, the evapotranspiration





calculation sub-model, the rainfall-runoff calculation sub-model, the confluence calculation
sub-model, the parameter sensitivity analysis and the parameter optimization sub-model. In
the vertical structure of the Liuxihe model, there are 3 layers from top to bottom: the canopy
layers, the soil layers and the underground layers, respectively. And the horizontal structure is
also divided into 3 types: the river cells, the hill slope cells and the reservoir cells. More
details of the Liuxihe model structure and its application effects can be found in the studies by
Chen (2009, 2018) and Li (2017, 2019).
4.2 Karst-Liuxihe model
The Liuxihe model prototype is a terrestrial hydrological mechanism model, which is
particularly useful in rainfall-runoff and confluence calculations, as the model performs well
in forecasting the river surface. To be suitable for karst basins, the structure of the Liuxihe
model should be improved to effectively adapt to the complex karst hydrogeological
conditions, which involves adding the karst mechanism to the model. A new distributed
hydrological model in this study, the Karst-Liuxihe model, was proposed on the prototype of
Liuxihe model to simulate and forecast the karst flood events. The process of improving the
structure of the Karst-Liuxihe model is summarized as follows.
1. Make the karst water-bearing media simplification in the model
In general, the karst hydrological process is hard to accurately forecast using a
hydrological model due to the complicated and anisotropic hydrogeological conditions of the
karst aquifers. Therefore, the water-bearing media in the karst aquifer must be effectively
simplified before building the model. First, the karst underground river system was
generalized into a multiple spatial structure in the model, where the water movement rules of
the underground river could be intelligible and computable. Second, the groundwater
movement patterns are divided into slow flow and rapid flow in the model. Slow flow mainly
exists in the tiny karst fissures, and rapid flow mainly occurs in wide karst cracks, conduits,
sinkholes and the underground river. Atkinson (1977) noted that when the width of the karst
fissure exceeds 10 cm, the water flow in the karst water-bearing medium is a non-Darcy flow,
i.e., turbulence with a rapid speed. The 10-cm width of the karst fissure was treated as a
threshold in this study, and when the width exceeded 10 cm, the groundwater movement
pattern was divided by the rapid flow. Otherwise, the flow was slow flow. In fact, a threshold
of 10 cm is sufficient in terms of contribution to flooding, especially for such a large study
area ($5.8 \times 10^4$ km$^2$).
2. Refine model structure and divide into KHRUs
The entire study area would be divided into a lot of grid cells by the high-resolution
DEM data, and these grid cells are known as karst sub-basins. The confluence path for each
karst sub-basin to the outlet of the basin is clear. Furthermore, to be suitable for the complex




karst aquifer and water-bearing media in karst basins, the model structure must be fine
enough to meet the flood simulation and forecasting requirements. Therefore, the karst
subbasins can be further divided into many KHRUs using GIS technology combined with the
karst landform in this paper, and the spatial variations in the karst subbasins can be subtly
described. Each KHRU had its own model parameters, and calculations of the entire karst
hydrological process, including calculations of precipitation, evapotranspiration, rainfall-
runoff and confluence, are independent of each other in each KHRU. This type of multiple
spatial structure in the model could effectively make maximum use of the limited
meteorological and hydrogeological data. In the vertical structure of the KHRU in the Karst-
Liuxihe model, there are 5 layers, including the vegetation cover, the soil layer, the epikarst
zone, the bedrock layer as well as the underground river. Water movement and exchange
rules between the karst fissure and conduit in the epikarst zone were reasonably considered in
this study. Fig. 8 shows the structure map of the KHRU.

a. The structure of the KHRU and the partial enlarged detail

b. A picture of the KHRU

Figure 8. The 3-dimensional spatial structure of the KHRU.

In Fig. 8, the partially enlarged details of Fig. 8a and b show the 3-dimensional spatial

model of the KHRU that is built in our laboratory, which is used to observe the slow and
rapid flows transfer into the karst fissures and conduits more intuitively. This process may be
necessary and helpful for modelling.
3. Increase the calculation of water movement rules in the karst aquifers

There is no module to address the water movement rules in the epikarst zone in the

Liuxihe model prototype. In the Karst-Liuxihe model in this study, the karst aquifer system
was divided into karst fissure and conduit systems, in which the water movement rule was
divided into slow flow and rapid flow. The 10-cm width of the karst crack is a threshold
(Atkinson, 1977); when the width exceeds 10 cm, the water movement pattern is divided by
the rapid flow. Otherwise, the flow is the slow flow. The karst fissure systems were mainly
the rock matrix and some small fissures, while the conduit systems include the wide fissures
and conduits as well as the karst shaft, sinkhole, and underground river during the floods. The
water movement was slow in the small karst fissure system and obeys Darcy's law. Therefore,
in the Karst-Liuxihe model, the system was generalized to an equivalent porous medium. A
3-dimensional equation of groundwater motion was used to describe the slow flow:
$$\frac{\partial}{\partial x}\left(K_{xx}\frac{\partial h}{\partial x}\right)+\frac{\partial}{\partial y}\left(K_{yy}\frac{\partial h}{\partial y}\right)+\frac{\partial}{\partial z}\left(K_{zz}\frac{\partial h}{\partial z}\right)\pm W=S_s\left(\frac{\partial h}{\partial t}\right) \tag{4}$$

where $K_{xx}$, $K_{yy}$, and $K_{zz}$ are the permeability coefficients of the rock mass in the *X, Y,* and



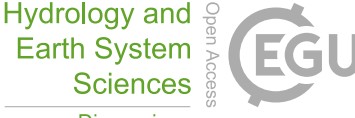

$Z$ directions, respectively, m d$^{-1}$; $h$ is the groundwater head, m; $W$ is the source-sink term, d$^{-1}$;
$S_s$ is the storage coefficient, m$^{-1}$; and $t$ is the time, d.
The conduit systems were generalized to multiple circular tubes, considering that the
tubes were mostly under pressure during the floods. Thus, the conduit systems were bearing
tubes in this paper. In these bearing tubes, when the groundwater was in a state of laminar
flow, the water flows of the tubes were calculated by the Hagen-Poiseuille equation:

$$Q = -A \frac{gd^2 \partial h}{32v \partial x} = -A \frac{\rho g d^2 \Delta h}{32 \mu \tau \Delta l} \qquad (5)$$

where $Q$ is the water flow of the laminar flow, m$^3$ s$^{-1}$; $A$ is the tube cross-sectional area, m$^2$; $d$
is the pipe diameter, m; $\rho$ is the density of the underground water, kg m$^{-3}$; $g$ is gravity
acceleration, m s$^{-2}$; $v=\mu/\rho$ is the coefficient of kinematic viscosity, and this value can be
calculated from the temperature (Shoemaker, 2008); $\partial h/\partial x=\Delta h/\tau \Delta l$ is the hydraulic slope
of the tubes, and $\tau$ is the tube curvature, which is a dimensionless parameter here.
When the groundwater was in a state of turbulent flow, the water flows of the tubes were
calculated by the Darcy-Weisbach equation:

$$Q = -2A \log \left( \frac{k_c}{3.71d} + \frac{2.51v}{d\sqrt{\frac{2gd\partial h_c}{\partial x}}} \right) \sqrt{\frac{2gd\partial h_c}{\partial x}}$$

$$= -2A \sqrt{\frac{2gd|\Delta h|}{\Delta l \tau}} \log \left( \frac{k_c}{3.71d} + \frac{2.51v}{d\sqrt{\frac{2gd^3|\Delta h|}{\Delta l \tau}}} \right) \frac{\Delta h}{|\Delta h|} \qquad (6)$$

where $Q$ is the water flow of the turbulent flow, m$^3$ s$^{-1}$; $f$ is the friction factor, dimensionless
here; $k_c$ is the average tube wall height, m; $R_e=Vd/v$ is the Reynolds Number, and $V$ is the
average velocity of the tubes, m s$^{-1}$. The Reynolds Number is divided into the upper Reynolds
Number and the lower Reynolds Number to determine whether the flow in the tubes is
laminar and turbulent. When there was laminar flow, the Reynolds Number at that time was
greater than the upper Reynolds Number. Then, the groundwater in the tubes transitioned
from laminar flow to turbulent flow. When there was turbulent flow, the Reynolds Number at
that time was less than the lower Reynolds Number, and the groundwater in the tubes
transitioned from turbulent flow to laminar flow.
In the unsaturated zone of the karst aquifer, there is usually an exchange of water
between slow flow and rapid flow, i.e., the exchange of water exists between each conduit





node and the connecting fissure node, and the exchange of water flow could be calculated
using this equation:
$$\begin{cases} Q = \alpha_{i,j,k}\left(h_n - h_{i,j,k}\right) \\ \alpha_{i,j,k} = \sum_{ip=1}^{np} \dfrac{\left(K_w\right)_{i,j,k}\,\pi d_{ip}\,\dfrac{1}{2}\left(\Delta l_{ip}\tau_{ip}\right)}{r_{ip}} \end{cases} \tag{7}$$

where $\alpha_{i,j,k}$ is the exchange coefficient at grid cell i, j, k of the KHRU, m$^2$ s$^{-1}$; $h_n$ is the head
value of the corresponding tube node, m; $h_{i,j,k}$ is the head value of the grid cell i, j, k, m; $np$
is the tube number than connected the i, j, k tube node; $\left(K_w\right)_{i,j,k}$ is the permeability
coefficient of the tube wall, m d$^{-1}$; $d_{ip}$ is the pipe diameter of tube $ip$, m; $\Delta l_{ip}$ is the length of
the connection between the i and p tube node, m; $\tau_{ip}$ is the tube curvature, and $r_{ip}$ is the tube
radius, m.
4. Add some necessary hydrogeological parameters to the model

In the original Liuxihe model, there are 14 parameters that require optimization, and

after adding the karst mechanism and especially by adding some necessary hydrogeological
parameters in the Karst-Liuxihe model. Then, the parameters were increased to 20, and
among them 18 need to be optimized. The remaining 2 parameters were the flow direction
and slope, which can be directly calculated from the high-resolution DEM data.

These added parameters could represent the underground water movement rules in the

epikarst zone and the underground river. The 6 added parameters are the macro crack volume
ratio, $V$; the permeability coefficient, $K$; the specific yield of the aquifer, $\chi$; thickness of the
karst aquifer, $h$; depletion coefficient, $\omega;$ and channel roughness, $n_1$. The parameters added
into the Karst-Liuxihe model will inevitably lead to uncertainties in the model during flood
simulation and forecasting, so the parameter sensitivity must be effectively analysed and
evaluated. In this study, a parameter sensitivity analysis method, known as the multiparameter
sensitivity analysis (MPSA) by Choi (1999) et al., was developed based on the Generalised
likelihood uncertainty estimation (GLUE) method to evaluate the parameter sensitivity in the
model.
**5. Coupled model set up**
5.1 Model setup

In general, there are many pits in the karst areas, and some of which are the false pits.



The existence of false pits is due to wrong data and systematic errors of DEM itself. These
false pits need to be reasonably filled before building the coupled model. Because there are
karst depressions and sinkholes in the karst areas, which cause true pits to exist, the model
retained these true pits, including the depressions and sinkholes. These true pits in the study
area play an important role in the flood transmission process and can be found through a field
survey. Due to the detention effect and peak clipping in the karst depressions, the
hydrological process is delayed, especially for the flood peak flow. This effect must be
considered in the coupled model, which can make a better performance for the model in karst
flood events simulations and forecasting. Before building the model, whether there exists a
detention effect and peak clipping in the karst depressions and sinkholes in the study area is a
key factor. If so, the storage capacity and size of these pits must be determined by a field
survey during floods. The capacity can be deduced according to the water level, and the
amounts of stranded floods near the pits must be considered in the water balance calculation
in the model. The specific calculation steps in the coupled model are shown below.
1. First, the limit discharge capacity of the underground river entrance in the study area, i.e.,
$Q_{max}$, was deduced through a field investigation and monitoring.
2. Then, the water inflow from the entrance of the underground river, i.e., $Q_{in}$, can be
calculated through the coupled model.
3. The relationship between $Q_{in}$ and $Q_{max}$ was compared to determine whether the flood
detention phenomenon was generated.
If $Q_{in} > Q_{max}$, the flood detention phenomenon is generated, and then, the flow of the
underground river outlet, $Q_{out} = Q_{max}$ is generated. The water storage of the flood detention
from the entrance of the underground river, $Q_s$, is as follows:

$$Q_s = Q_{s1} + Q_{in} - Q_{max} \tag{8}$$

where $Q_s$ is the water storage of the flood detention during this period, $m^3 \ s^{-1}$; $Q_{s1}$ is the water
storage of the flood detention from the preceding time period, $m^3 \ s^{-1}$; and if there is no flood
detention, i.e., $Q_{s1}=0$.
If $Q_{in} \leq Q_{max}$, and $Q_{s1}=0$, then

$$Q_{out} = Q_{in} \tag{9}$$

If $Q_{in} \leq Q_{max}$, $Q_{s1}>0$, and $Q_{in} + Q_{s1} \leq Q_{max}$, then

$$Q_{out} = Q_{in} + Q_{s1} \tag{10}$$

Otherwise, if $Q_{in} \leq Q_{max}$, $Q_{s1}>0$, and $Q_{in} + Q_{s1} > Q_{max}$, then

$$Q_{out} = Q_{max} \tag{11}$$





In this study, the entire karst basin was divided into 1,469,900 KHRUs in the Karst-
Liuxihe model using the 200 m×200 m high-resolution DEM data. There were 6,696 river
cells and 1,463,204 hill slope cells. The river system was divided into a 4-order stream based
on Strahler's method, which is shown in Fig. 1a. The KHRU in the coupled model (Fig. 8),
which is the smallest unit, was proposed to effectively reflect the complicated
hydrogeological condition of the underlying surface and karst aquifers. All the hydrological
processes, including evapotranspiration and rainfall-runoff, confluence as well as the
parameter optimization, were calculated on this KHRU and because the KHRU was
completely physically-based, the differences in the complex hydrogeological characteristics
of karst aquifers could be truly reflected. Therefore, the model effect and performance in karst
forecasting could be reliably improved in this way.
After division of the KHRUs, i.e., model setup was finished, the postprocessed WRF QPF
and the PERSIANN-CCS QPEs results were fed into the Karst-Liuxihe model to validate its
feasibility in karst floods simulations and forecasting.
5.2 Parameter optimization
There are 20 parameters in the Karst-Liuxihe model, and among these parameters, 18
needed to be optimized. In this study, an improved PSO algorithm, mainly the algorithm
parameters, were revised to improve the performance and convergence efficiency (Chen et al.,
2016); this improvement can largely improve the accuracy of the coupled model in flood
simulations and forecasting in a karst basin. The observed rainfall and karst flood event data
as well as the hydrogeological data of the karst underlying the surface and aquifer were
adopted to optimize the parameters of the Karst-Liuxihe model in this paper. These data were
fully physically-based that can describe the complex karst water-bearing medium effectively.
There are 30 floods in the study area from 1982-2013, which were used to verify the
model effect in the karst hydrological processes simulations and prediction. The flood
prediction results were very good (Li et al., 2019), implied that the model can be effectively
applied in karst areas. In this study, 8 karst flood events, including floods 2005061400,
2006060400, 2007070800, 2008060900, 200906090800, 201106010900, 201206022000 and
201306011400, were used to test the coupled model performance in the karst floods
forecasting, i.e., coupling the Karst-Liuxihe model with the 2 weather models, WRF QPF and
PERSIANN-CCS QPEs. Among these flood events, floods 2005061400, 2006060400,
2007070800 and 2008060900 were used for parameter optimization, and the best flood
simulation based on these four floods was used for the final parameter optimization. The
remainder of the floods were adopted for model validation. The parameter evolution results of
the coupled model are shown in Fig. 9.
Figure 9. The parameter evolution results.





**6 Results and discussion**
6.1 Results of the parameter optimization
From the parameter evolution results in Fig. 9, the parameter evolution process began very
volatile, and after a few cycles, approximately 20 times, the evolution leveled off and held
steady after 40 cycles, which signified that the parameter optimization had converged. The
thickness of a lines in Fig. 9 indicates the sensitivity of the parameters, and the thicker the
line is, the more sensitive the parameter will be. The sensitivity of the parameters will be
elaborated upon in the next section of the paper (section 6.2). The karst floods simulation
effects based on parameter optimization were drawn in Fig. 10, and the evaluation indices of
the flood simulations were listed in Table 2.
Figure 10. The karst floods simulation effects of the coupled model.
Table 2. Evaluation indices for the karst floods simulation effects.
From Fig. 10, the karst flood simulated effect of flood 2008060900 was the best, especially
for the simulated flood peak flow, was the closest to observed peak flow. To further compare
the effects of the flood simulations, the 6 evaluation indices, including the Nash-Sutcliffe
coefficient/C; the coefficient of the water balance/W; the correlation coefficient/R; the flood
peak flow relative error/E%; the process relative error/P% as well as the flood peak flow time
error/T(hours), were listed in Table 2. These indices were also the best for modelling flood
2008060900. Therefore, flood 2008060900 was finally used for the parameter optimization.
The reasonable simulated flood processes based on the improved PSO algorithm for the
coupled model were suited the practically observed values very well (as shown in Fig. 10 and
Table 2), which implied that the parametric optimization method in this study, i.e., the
improved PSO algorithm was feasible and effective.
6.2 Model uncertainty analysis
The uncertainty analysis of the coupled model in this study could be effectively solved
with 3 aspects: 1. Ensure the reliability of the model input data, which include rainfall data,
karst flood events, and hydrogeological data. Among these data, the rainfall data can be
reliably obtained by WRF QPF and PERSIANN-CCS QPEs; the karst flood events were
obtained from the local hydrology department, and the hydrogeological data were obtained
through a field survey and tracer testing in the study area. 2. Solve the uncertainty problem of
model structure through model structure and function improvement (as shown in section 4.2).
3. Solve the uncertainty problem of the model parameters.
The uncertainty analysis of the parameters for the coupled model mainly means the
parameters sensitivity analysis in this study. The sensitivity analysis method used in this
paper, which is known as MPSA (Choi et al., 1999), was improved on the Generalized
Likelihood Uncertainty Estimation (GLUE) algorithm. The Nash-Sutcliffe coefficient/C, as

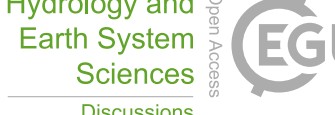



the objective function, was used to analyse the sensitivity of the coupled model parameters in
this study, the equation of the objective function was as follows:
$$NSE = 1 - \frac{\sum_{i=1}^{n}\left(Q_i - Q_i^{'}\right)^2}{\sum_{i=1}^{n}\left(Q_i - \overline{Q}\right)^2}$$
(12)

where $NSE$ was the value of the objective function, i.e., Nash-Sutcliffe coefficient/C; $Q_i$
and $Q_i^{'}$ were observed and simulated water flows, respectively, $m^3 s^{-1}$, $\overline{Q}$ was the average
observed water flow value, $m^3 s^{-1}$, and $n$ was the observed period numbers, hours.

Table 3 shows the results of the parameters sensitivity calculation. In Table 3, the closer

the value of the objective function for the parameter is to 1, the more sensitive the parameter
will be.
Table 3. The calculation results of the coupled model parameters sensitivity.

From the results shown in Table 3, the value of the objective function for the parameter-

saturated water content, $\theta_{sat}$ was the maximum one. This means that the parameter, $\theta_{sat}$ is the
most sensitive parameter of the Karst-Liuxihe model. The parameter sensitivity is also shown
in Fig. 9. The thickness of the line in Fig. 9 indicates the parameter sensitivity, and the thicker
the line, the more sensitive the parameter will be, which can represent the sensitivity of
parameters more intuitively. From Table 4 and Fig. 9, the sequence of parameter sensitivity
of the Karst-Liuxihe model was as follows: $\theta_{sat} > \theta_s > \theta_{fc} > K_s > V > K > \chi > h > z > b > S_w >$
$S_p > n > n_1 > \omega > \lambda > E_p > C_{wl}$. The name of these parameters are shown in Table 3.
6.3 Floods simulations with the postprocessed 2 weather models

In this study, to analyse the effects of the karst flood simulation using the initial WRF

QPF, the PERSIANN-CCS QPEs and their postprocessed results, the karst flood events,
floods from 2008-2013 were simulated by the coupled model. The results comparisons are
shown in Fig. 11 to Fig. 15.
Figure 11. The flood simulation results of flood 2008060900 based on the coupled model. (a)
is the postprocessed WRF flood simulation result, and (b) is the postprocessed PERSIANN-
CCS flood simulation result.
Figure 12. The flood simulation results of flood 200906090800 based on the coupled model.
(a) is the postprocessed WRF flood simulation result, and (b) is the postprocessed
PERSIANN-CCS flood simulation result.
Figure 13. The flood simulation results of flood 201106010900 based on the coupled model.
(a) is the postprocessed WRF flood simulation result, and (b) is the postprocessed
PERSIANN-CCS flood simulation result.





Figure 14. The flood simulation results of flood 201206022000 based on the coupled model.
(a) is the postprocessed WRF flood simulation result, and (b) is the postprocessed
PERSIANN-CCS flood simulation result.
Figure 15. The flood simulation results of flood 201306011400 based on the coupled model.
(a) is the postprocessed WRF flood simulation result, and (b) is the postprocessed
PERSIANN-CCS flood simulation result.
From Fig. 11 to Fig. 15, the floods simulations with the original WRF QPF and
PERSIANN-CCS QPEs products were unsatisfactory, especially for the simulated peak flows.
In contrast, the coupled model performance with the postprocessed WRF QPF and
PERSIANN-CCS QPEs were better. The simulated flood peak errors of the postprocessed
weather models were effectively reduced. For further comparison, the 6 evaluation indices of
the floods simulations with the original weather models and the postprocessed models are
shown in Table 4.
Table 4. The evaluation indices of karst floods simulations with the original WRF QPF and
PERSIANN-CCS QPEs and their postprocessed values.
From Table 4, all of these 6 evaluation indices with the postprocessed WRF QPF and
PERSIANN-CCS QPEs had improved than those with the original 2 weather models. For the
WRF QPF, after postprocessing, the average water balance coefficient increased by 8%;the
average Nash-Sutcliffe coefficient increased by 3%; and the average correlation coefficient
increased by 2%. While the average process relative error decreased by 5%; the average peak
flow relative error decreased by 5%;and the peak flow time error decreased by 2 hours,
respectively. For the postprocessing PERSIANN-CCS QPEs, the average Nash-Sutcliffe
coefficient increased by 5%; the average water balance coefficient increased by 4%; and the
average correlation coefficient increased by 4%; While the average process relative error
decreased by 5%; the average peak flow relative error decreased by 6%; and the average peak
flow time error decreased by 3 hours, respectively. Obviously these evaluation indices were
getting better following postprocessing of WRF QPF and PERSIANN-CCS QPEs, which
implied that the postprocessing method for the 2 weather models in this study was effective
and reasonable.
6.4 Verify the coupled model performance by comparing 3 kinds of precipitation
products
There are 3 kinds of precipitation products that are used in this study, i.e., rain gauge
precipitation, postprocessed WRF QPF and postprocessed PERSIANN-CCS QPEs. The
effects of different types of precipitation products on the flood process simulated by
hydrological model are calculated and compared to test their performance. The flood events
included floods from 2008-2013, which were simulated by the coupled model. The results
comparison is shown in Fig. 16, and Table 5.



Figure 16. The karst floods simulated effects of the coupled model with the 3 precipitation
products.
Table 5. The evaluation indices of karst floods simulations with the 3 precipitation products.

From Fig. 16 and Table 5, the flood processes simulated by the Karst-Liuxihe model

using the rain gauge precipitation were better than those of the postprocessed WRF QPF and
PERSIANN-CCS QPEs. The rain gauge precipitation can directly reflect the actual rainfall
situation in the basin, which is the reason that the rain gauge precipitation, taken as the true
value, was used to calibrate the weather models in this paper. However, this kind of
precipitation based on rain gauge measurements has no lead time because the rain has fallen
to the ground. In addition, there is usually a shortage of rain gauges in karst areas. Therefore,
the WRF QPF and the PERSIANN-CCS QPEs were adopted to obtain the effective
precipitation in the study area. From Fig. 16 and Table 5, compared with the karst flood
processes simulated with the postprocessed WRF QPF, the flood simulated results with the
postprocessed PERSIANN-CCS QPEs were slightly better. In particular, the peak flow
simulation demonstrated the superiority of the postprocessed PERSIANN-CCS QPEs.
However, the rainfall estimation results from PERSIANN-CCS have no lead time, while the
WRF QPF can offer rainfall forecasting with a lead time of 96 hours, which means that there
is a lead time of 96 hours for flood forecasting by coupling the Karst-Liuxihe model with the
WRF QPF. This lead time of the coupled model can provide more responses time for floods
warnings.

The satisfying flood simulated results in Fig. 16 and their rational evaluation indices in

Table 5 proved that coupling the 2 weather models with the Karst-Liuxihe model in this paper
was feasible and effective for the Liujiang basin. In particular, the flood detention and peak
clipping effect of the upstream karst depressions were considered in the coupled model
calculation, making the water balance calculation in the model more reasonable and reflecting
the actual flood evolution process in the karst area; the average coefficients of water balance/
W for the precipitation by rain gauges, WRF QPF and PERSIANN-CCS QPEs were 0.92,
1.07, and 0.89,  respectively (as shown in Table 5).  The water amount is basically balanced in
the model. Furthermore, the flood detention effect made the flood peak appear later in reality,
and by contrast, the simulated peak flow time came earlier, despite the flood detention effect
being considered in the model. The average peak time error, T for the rain gauge precipitation,
WRF QPF and PERSIANN-CCS QPEs were -5, -6, and -4,  respectively. In some ways, these
results provide an extra amount of lead time for flood forecasting. The peak clipping effect
considered in the coupled model brought the simulated peak flow value closer to that of the
observed value. The average peak flow relative error, E for the rain gauge precipitation, WRF
QPF and PERSIANN-CCS QPEs were 4%, 12%, and 8%,  respectively (as shown in Table 5).


Therefore, coupling the Karst-Liuxihe model with the postprocessed WRF QPF and
PERSIANN-CCS QPEs could largely improve the precision of the karst flood simulations
and forecasting.
**7 Conclusion**

The precipitation result, as a hydrological model input data, is one of the driving factors
that makes the model work swimmingly. However, it is often hard to acquire effective rainfall
results in karst areas. In this paper, WRF QPF and PERSIANN-CCS QPEs were adopted to
obtain acceptable precipitation results for the Liujiang karst river basin. A postprocessed
method was proposed to revise the rainfall products using these 2 weather models. To test the
effectiveness of this revision, the Karst-Liuxihe model was coupled with the postprocessed
WRF QPF and PERSIANN-CCS QPEs to simulate the floods of Liujiang karst watershed.
The Karst-Liuxihe model proposed in this study performed well in the flood simulations and
forecasting. The model structure and function was improved from various aspects, including
refining the model structure by putting forward the KHRUs in the model, increasing the
calculations of water movement rules in the epikarst zone and underground river, and by
adding some necessary hydrogeological parameters to the coupled model to reflect the true
conditions of rainfall-runoff in the karst underlying surface. The reasonable flood events
simulated effects by the improved Karst-Liuxihe model proved that the postprocessed method
proposed to revise the weather models in this paper was feasible. The following conclusions
were obtained from the study results of this paper.
1. The quantitative precipitation results produced by WRF QPF and PERSIANN-CCS QPEs
were quite closed to the observed rainfall data by rain gauge, especially in the rainfall
distribution. However, there is a relative error between the precipitation of the weather
models and the rain gauge, which was 17% with WRF QPF and -14% with PERSIANN-CCS
QPEs. This finding implied that WRF QPF overestimated the precipitation value, while
PERSIANN-CCS QPEs underestimated the precipitation values. The postprocessing method
proposed in this study could largely reduce these relative errors.
2. The model parametric uncertainty analysis showed that the parameter-saturated water
content, $\theta_{sat}$ was the most sensitive. The parameter sensitivity sequence of the Karst-Liuxihe
model was: $\theta_{sat} > \theta_s > \theta_{fc} > K_s > V > K > \chi > h > z > b > S_w > S_p > n > n_1 > \omega > \lambda > E_p > C_{wl}$.
3. Compared with the karst floods events simulated effects based on the initial 2 weather
models, the floods simulations with the postprocessed WRF QPF and PERSIANN-CCS QPEs
were much better. For the postprocessed WRF QPF, the average water balance coefficient,
Nash-Sutcliffe coefficient, and correlation coefficient were increased by 8%,3%,2%,
respectively. While the average peak flow relative error, process relative error, and the peak
flow time error were decreased by 5%,5%,2 hours, respectively. For the postprocessed



PERSIANN-CCS QPEs, the average water balance coefficient, Nash-Sutcliffe coefficient,
and correlation coefficient were increased by 4%,5%,4%, respectively. While the average
peak flow relative error, process relative error, and the peak flow time error were decreased
by 6%,5%,3 hours, respectively. It was obvious that the postprocessed method proposed in
this study was effective and feasible.
4. The flood processes simulated by the Karst-Liuxihe model using the rain gauge
precipitation were the best. Compared with the simulated floods with the postprocessed WRF
QPF, the simulation effects with the postprocessed PERSIANN-CCS QPEs were slightly
better, especially in the peak flow simulation. However, the rainfall data by the PERSIANN-
CCS QPEs had no lead time, which was applicable to the simulation and inversion after the
occurrence of floods. However, coupling the Karst-Liuxihe model with the WRF QPF model
resulted in a lead time of 96 hours in the flood forecasting, which can provide an adequate
amount of time for flood warnings and emergency responses. The satisfying flood simulated
results proved that coupling the 2 weather models with the Karst-Liuxihe model in this paper
was feasible and reasonable for the Liujiang karst river basin.
5. The flood detention and peak clipping effect of the upstream karst depressions were
calculated in the coupled model, which enabled the model to reflect the actual flood evolution
processes in the study area. The simulated average coefficients of water balance/W for the
observed precipitation by rain gauge, WRF QPF and PERSIANN-CCS QPEs were 0.92, 1.07,
and 0.89, respectively. The simulated average peak time error, T for the rain gauge
precipitation, WRF QPF and PERSIANN-CCS QPEs were -5, -6, and -4, respectively, and in
a way, provided extra lead time for the flood warning and forecasting. The simulated average
value of the peak flow relative error, E for the rain gauge precipitation, WRF QPF and
PERSIANN-CCS QPEs were 4%, 12%, and 8%, respectively, which were close to that of the
observation values. These results proved that coupling the Karst-Liuxihe model with the
postprocessed WRF QPF and PERSIANN-CCS QPEs in this paper could largely improve the
precision of karst floods simulations and forecasting. This coupled model could be effectively
adopted in other karst areas like Liujiang karst basin.

## 817 **Data availability.**

The observed rainfall data and the karst flood events are offered by Liuzhou hydrological bureau,
Guangxi province, China.
The WRF model for this study is the WRF-ARW model version 3.4; and the PERSIANN-CCS
QPEs data can be downloaded at no cost from http://cics.umd.edu/ipwg/us_web.html, last accessed: 18
March 2019. The Liuxihe model prototype is offered by Y, Chen (Chen, 2009).
The property data of the study area, including the DEM data, the land use type and the soil type,
can be downloaded at no cost. The DEM data are from http://srtm.csi.cgiar.org, last accessed: 02 April



2019. Land use types can be downloaded from http://landcover.usgs.gov, last accessed: 02 April 2019.
The soil types are from http://www.isric.org, last accessed: 05 April 2019.
**Author contributions.** The first and corresponding author is JIL, was in charge of the entire
paper, such as the model calculation and the writing of this paper and so on. DY provided advice on the
scientific issues raised in this article. YJ helped to conceive the structure of the model. JL provided
significant assistance in the English translation of the paper. YC offered the prototype of the Liuxihe
model.
**Competing interests.**
The authors declare that they have no conflicts of interest.
**Acknowledgments.** This study is supported by the National Key Research and Development
Program of China (2016YFC0502306), China Postdoctoral Science Foundation (2019M653316), the
Fundamental Research Funds for the Central Universities (XDJK2019C017), the Chongqing Municipal
Science and Technology Commission Fellowship Fund (No. cstc2018jcyj-yszx0013), the Open Project
Program of the Chongqing Key Laboratory of Karst Environment (Grant No. Cqk201801), and the
Open Project Program of the Laboratory of Chongqing groundwater resourse utilization and
environmental protection.





**Figures**

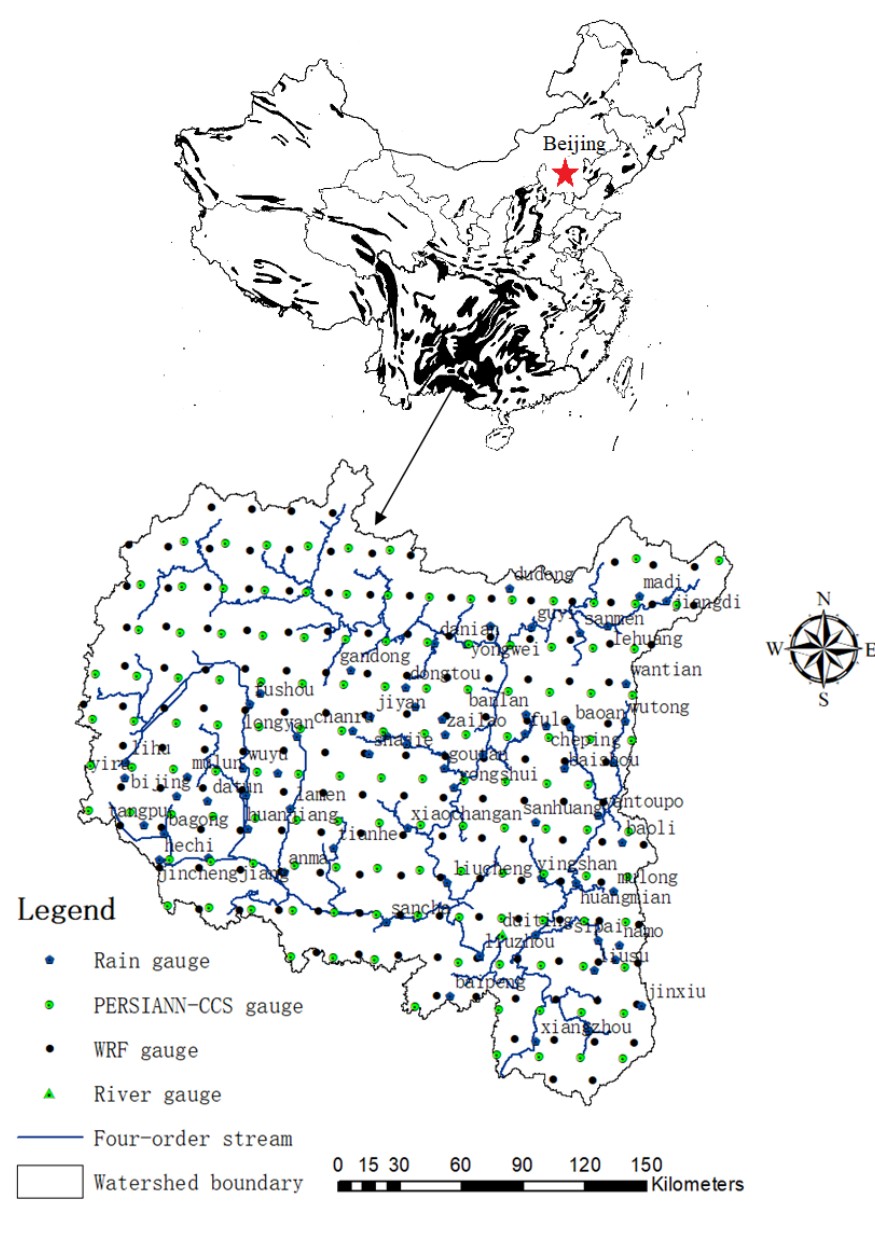


a. Gauges map (The black patches of the map are karst areas)



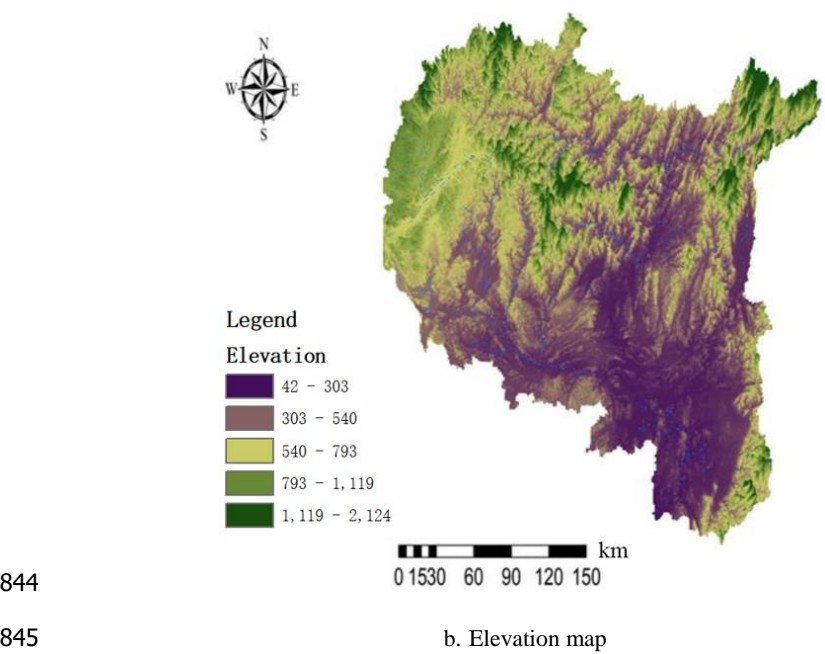


b. Elevation map

Figure 1. The sketch map of Liujiang karst watershed.

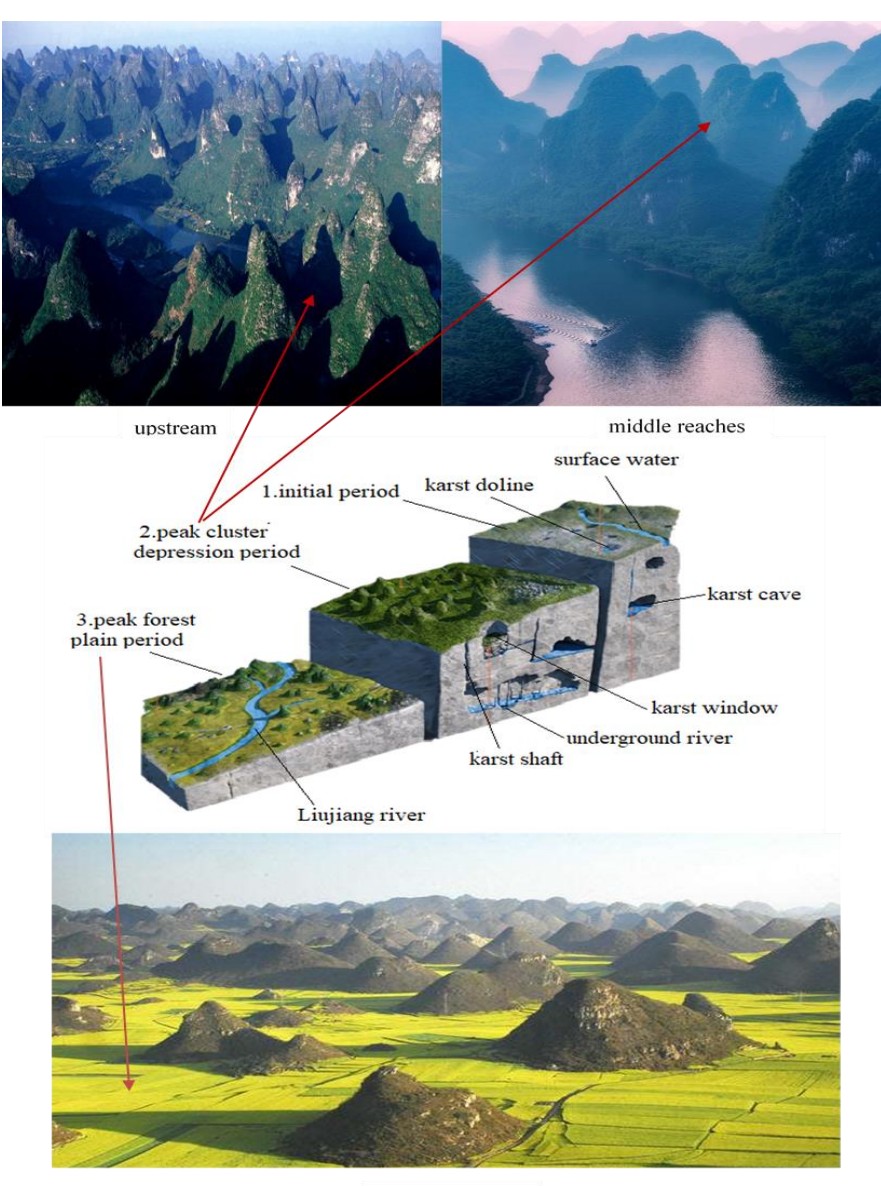


Figure 2. The karst landform evolution of the Liujiang basin(The photographs of the basin
upstream is from http://guilinkarst.com/en/nd.jsp?id=113, last access:10 April 2019. The
photographs of the middle reaches is captured by planet institute at
https://mp.weixin.qq.com/s?mpshare=1&scene=22&mid=2247521167&sn=
a3bf8521fda8e297ed58eae7e07bdc67&idx=1&__biz=MzIyOTQ1OTYzMw%3D%3D&chks
m=e8408051df3709477da49ef4362bf2f40db5279360c32f118575b71d596af78d098beea814d
8&srcid=0402YfBsf64zXrtsVpHoAuHg#rd, last access:2 April 2019. And the photographs of
the basin downstream is from http://travel.sohu.com/
20130221/n366552284_2.shtml, last access:10 April 2019).





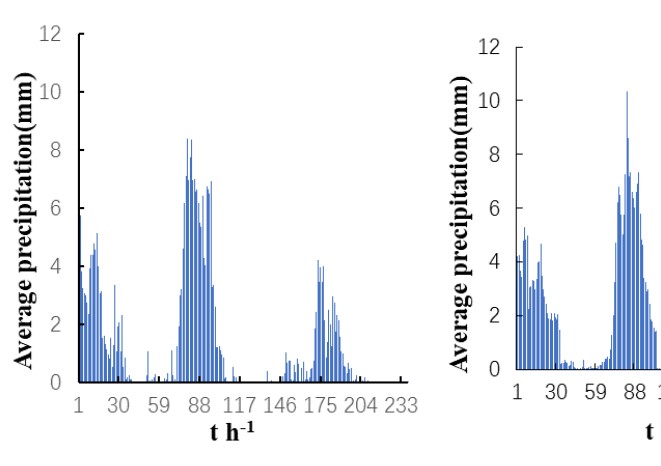


a.   Rain gauge precipitation              b. WRF QPF

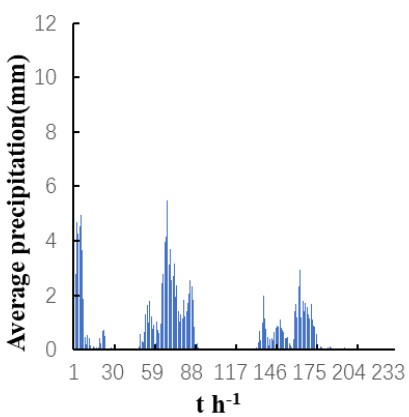


c.   PERSIANN-CCS QPEs

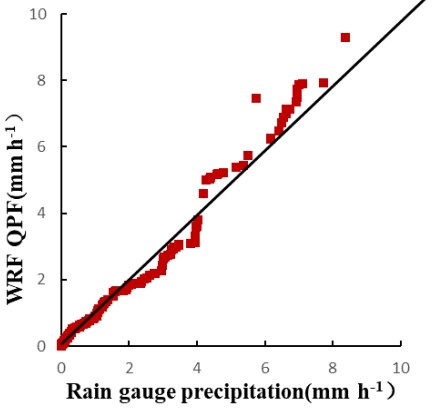

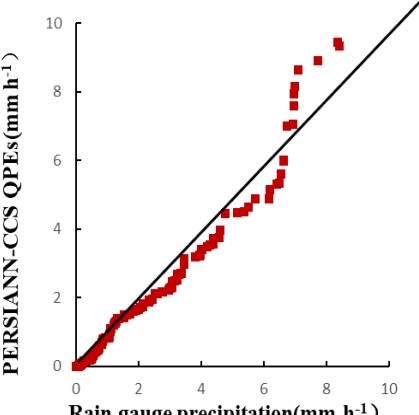






d. Quantile–quantile plot of WRF QPF     e. Quantile–quantile plot of PERSIANN-
and Rain gauge precipitation           CCS QPEs and Rain gauge precipitation
Figure 3. The rainfall results of the 3 precipitation products (2008).

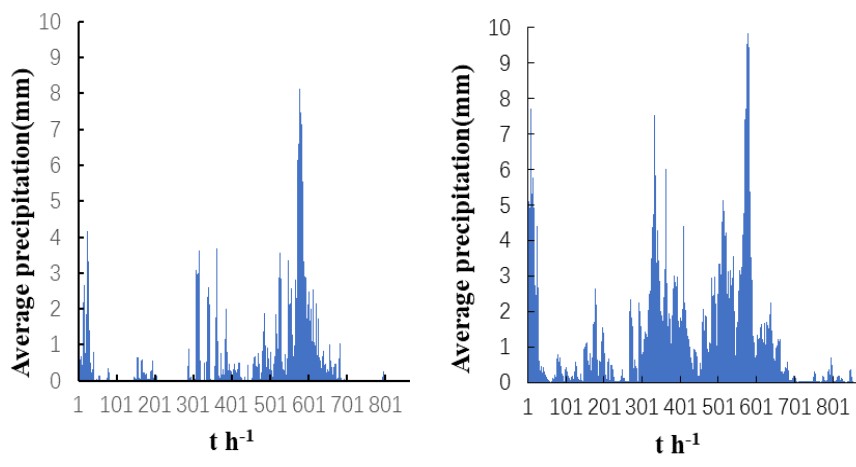


a.    Rain gauge precipitation            b. WRF QPF

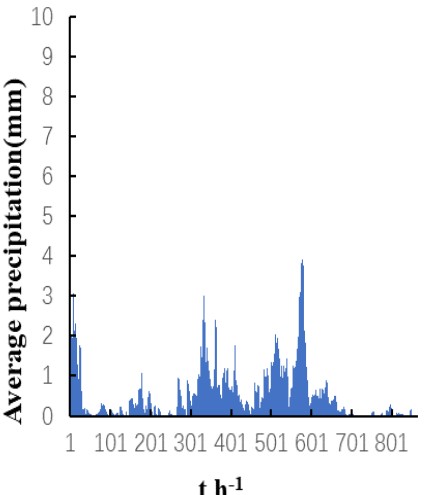


c.    PERSIANN-CCS QPEs


d. Quantile–quantile plot of WRF QPF      e. Quantile–quantile plot of PERSIANN-
and Rain gauge precipitation              CCS QPEs and Rain gauge precipitation
Figure 4. The rainfall results of the 3 precipitation products (2009).

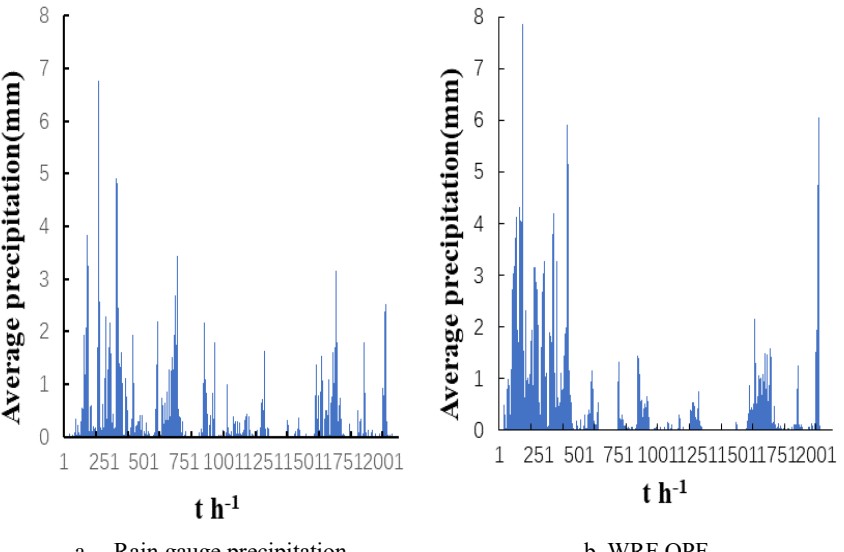

a.   Rain gauge precipitation                         b. WRF QPF



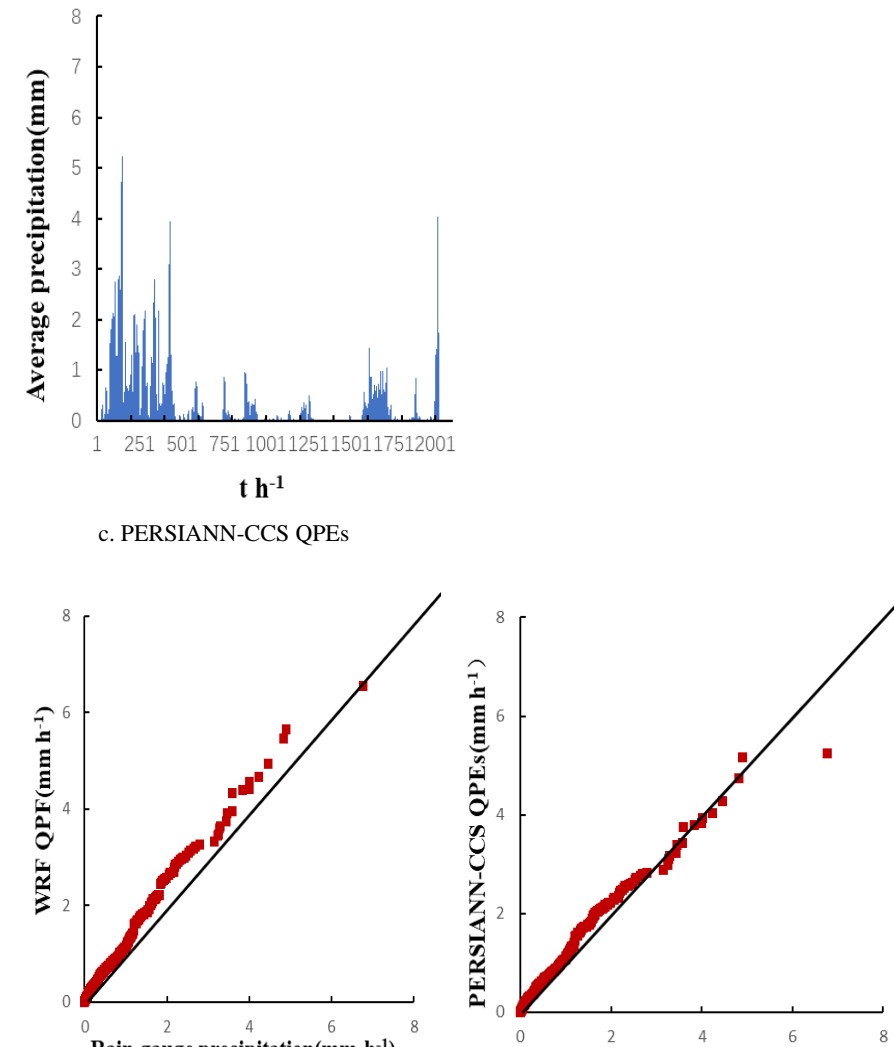


876    c. PERSIANN-CCS QPEs



879  d. Quantile–quantile plot of WRF QPF  e. Quantile–quantile plot of PERSIANN-

880   and Rain gauge precipitation     CCS QPEs and Rain gauge precipitation

881  Figure 5. The rainfall results of the 3 precipitation products (2011).





a.   Rain gauge precipitation              b. WRF QPF
c.   PERSIANN-CCS QPEs





d. Quantile–quantile plot of WRF QPF  e. Quantile–quantile plot of PERSIANN-
and Rain gauge precipitation     CCS QPEs and Rain gauge precipitation
Figure 6. The rainfall results of the 3 precipitation products (2012).

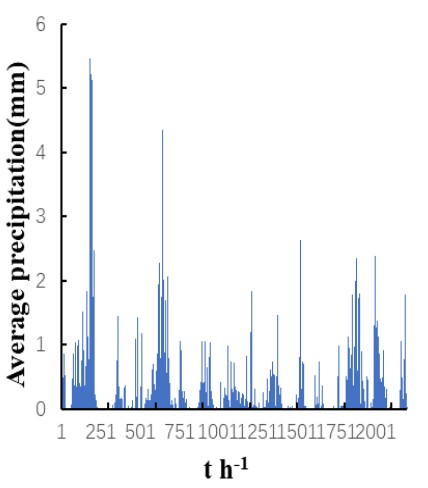

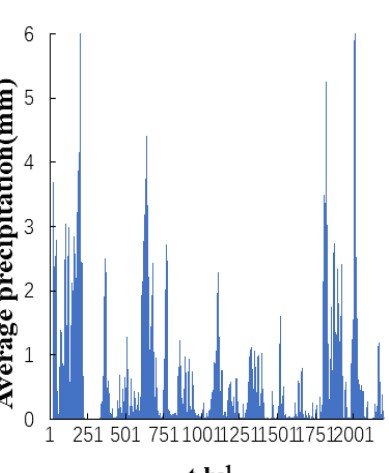


a. Rain gauge precipitation       b. WRF

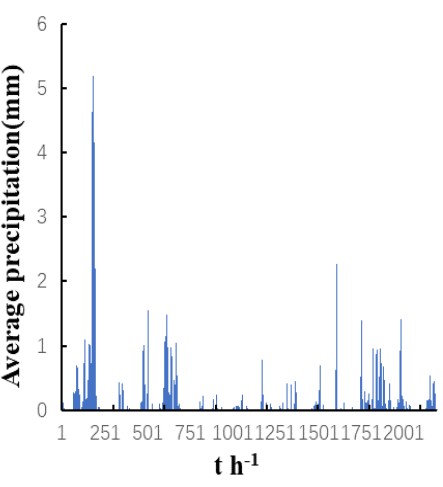


c. PERSIANN-CCS QPEs



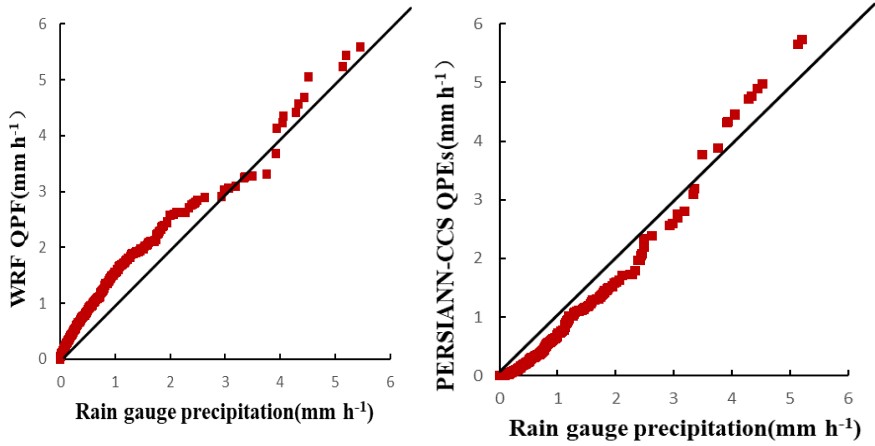

d. Quantile–quantile plot of WRF QPF        e. Quantile–quantile plot of PERSIANN-

 and Rain gauge precipitation              CCS QPEs and Rain gauge precipitation

Figure 7. The rainfall results of the 3 precipitation products (2013).

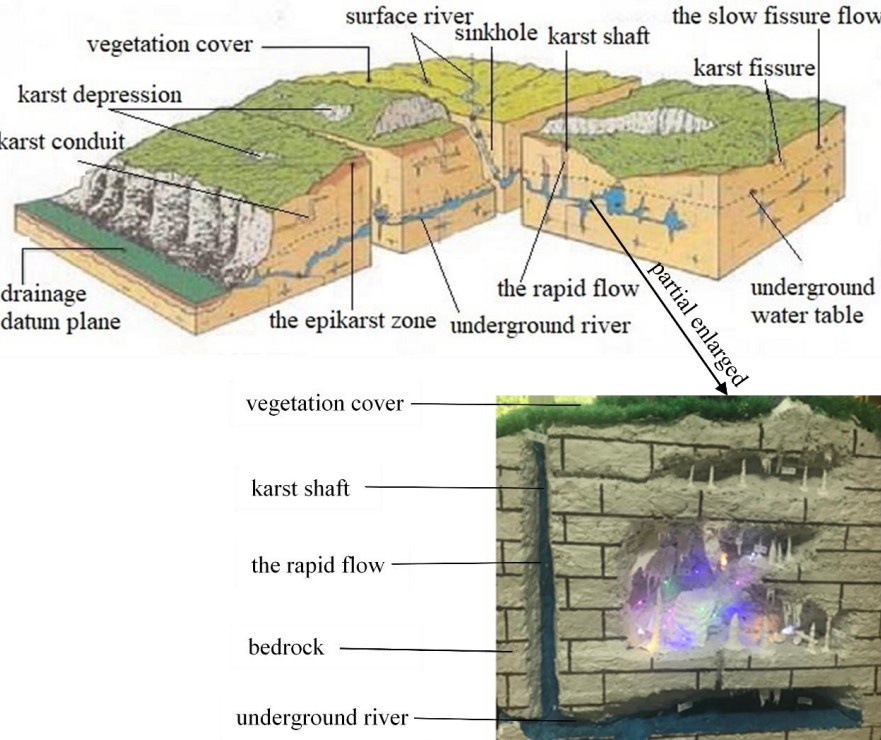

a. The structure of the KHRU and the partial enlarged detail




b. A picture of the KHRU

Figure 8. The 3-dimensional spatial structure of the KHRU.

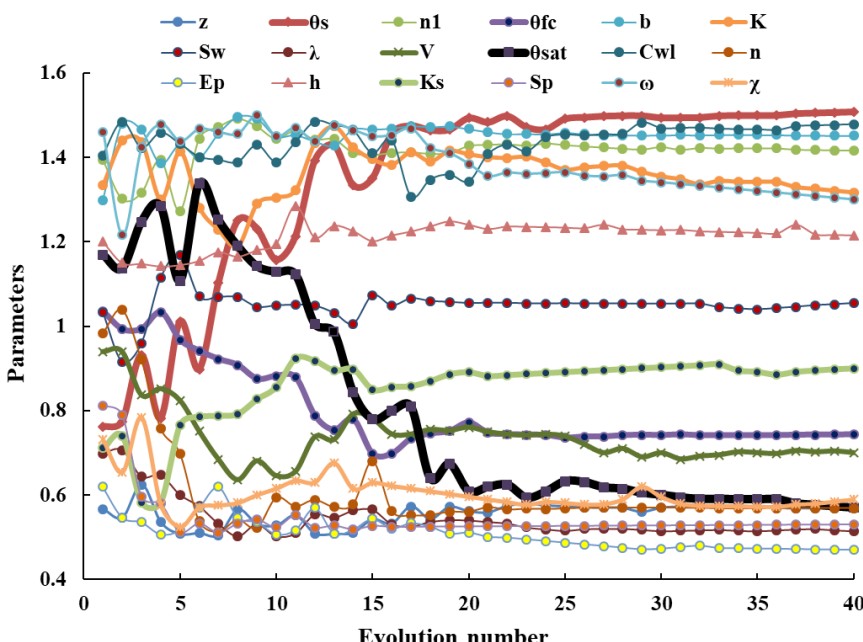


Figure 9. The parameter evolution results.



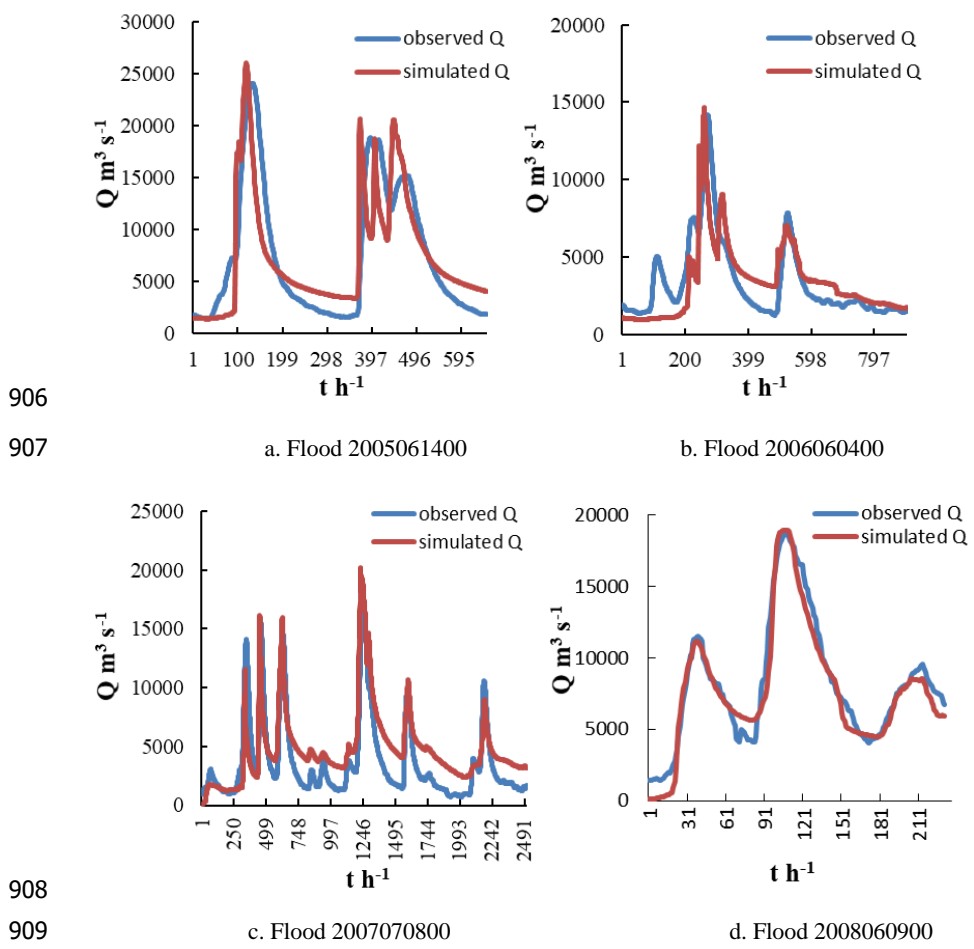


a. Flood 2005061400                    b. Flood 2006060400


c. Flood 2007070800                    d. Flood 2008060900

Figure 10. The karst floods simulation effects of the coupled model.

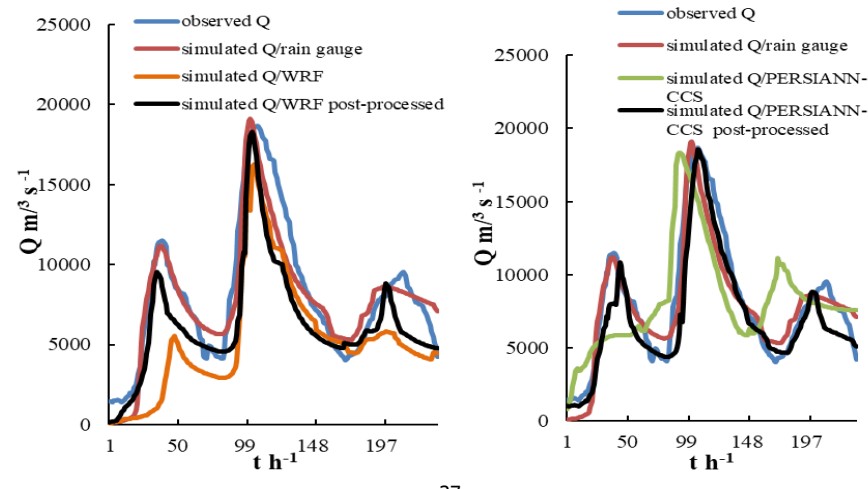






(a)                                    (b)

Figure 11. The flood simulation results of flood 2008060900 based on the coupled model. (a)
is the postprocessed WRF flood simulation result, and (b) is the postprocessed PERSIANN-
CCS flood simulation result.

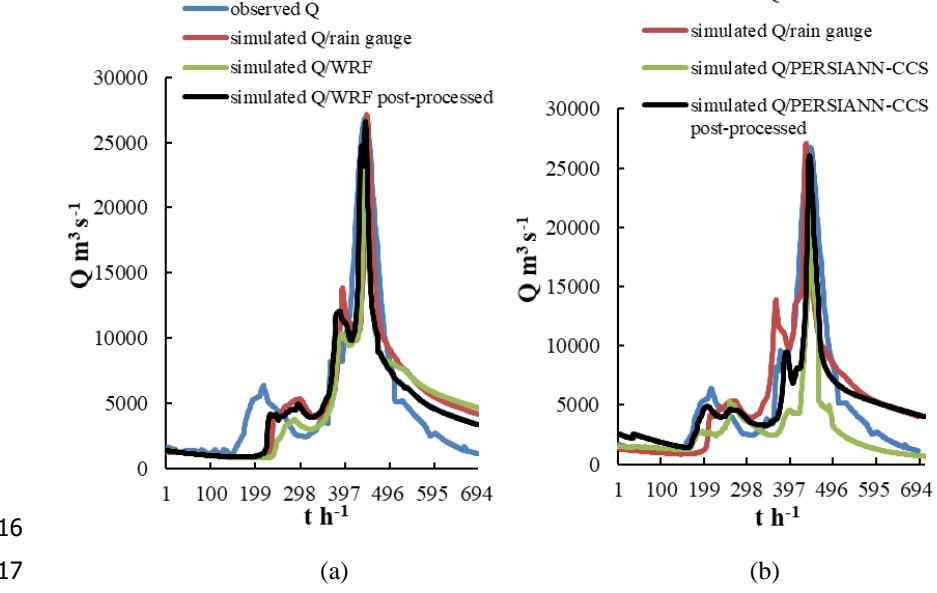


(a)                                    (b)

Figure 12. The flood simulation results of flood 200906090800 based on the coupled model.
(a) is the postprocessed WRF flood simulation result, and (b) is the postprocessed
PERSIANN-CCS flood simulation result.

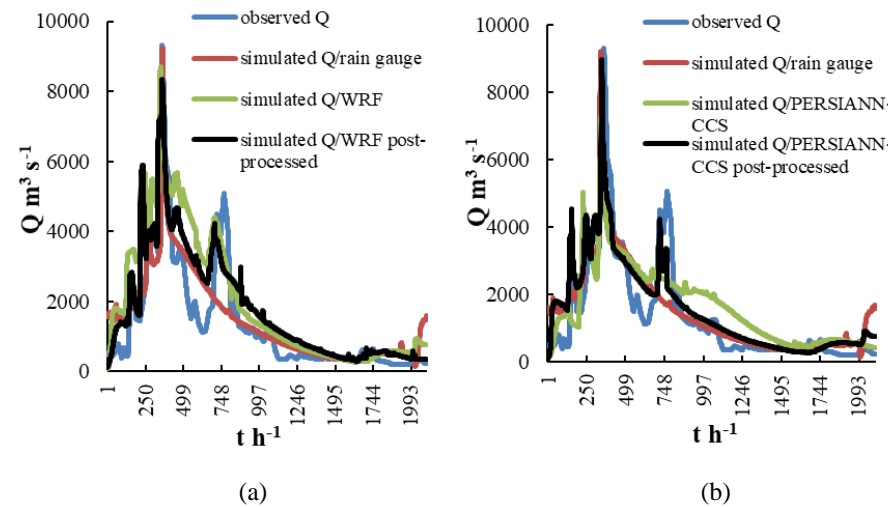


(a)                                    (b)





Figure 13. The flood simulation results of flood 201106010900 based on the coupled model.
(a) is the postprocessed WRF flood simulation result, and (b) is the postprocessed
PERSIANN-CCS flood simulation result.

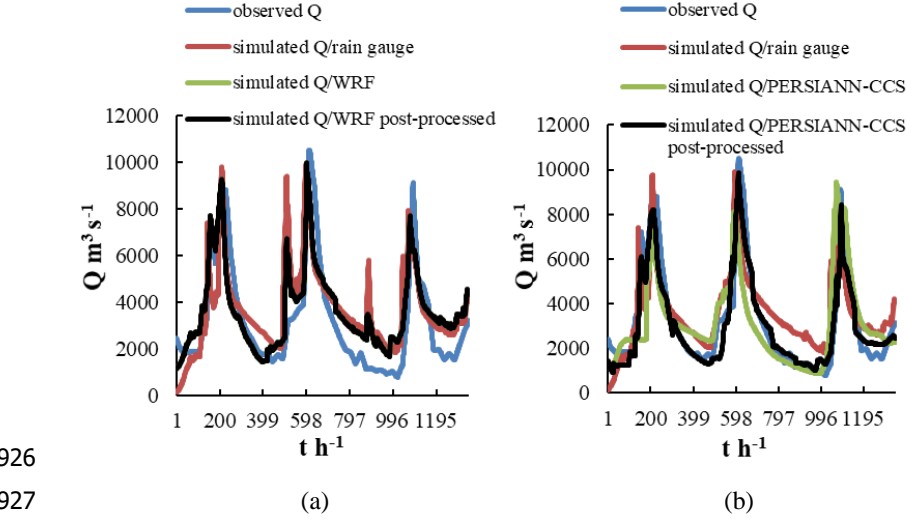


(a)                                              (b)

Figure 14. The flood simulation results of flood 201206022000 based on the coupled model.
(a) is the postprocessed WRF flood simulation result, and (b) is the postprocessed
PERSIANN-CCS flood simulation result.

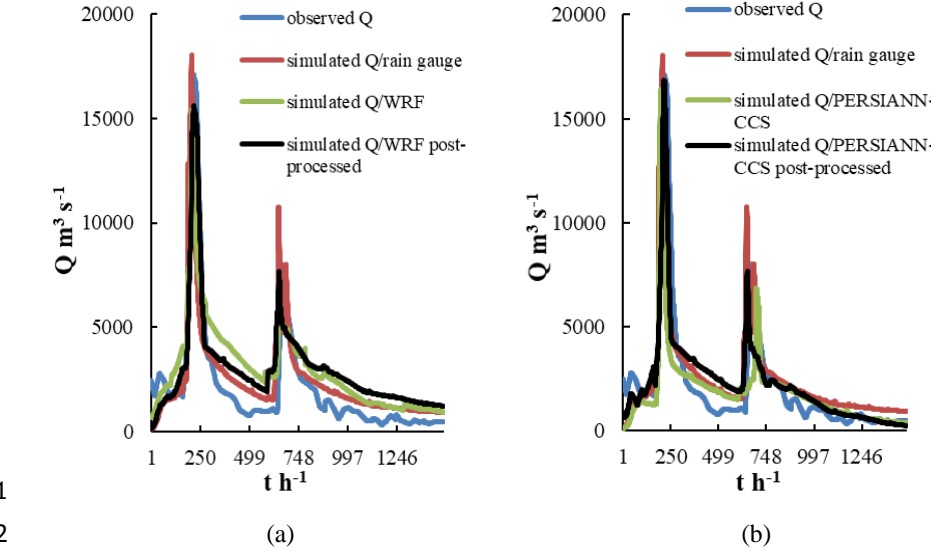


(a)                                              (b)

Figure 15. The flood simulation results of flood 201306011400 based on the coupled model.
(a) is the postprocessed WRF flood simulation result, and (b) is the postprocessed
PERSIANN-CCS flood simulation result.



      (a)    flood 2008060900           (b) flood 200906090800

      (c) flood 201106010900           (d) flood 201206022000

      (e) flood 201306011400

Figure 16. The karst floods simulated effects of the coupled model with the 3 precipitation
products.





## Tables

Table 1. The quantitative rainfall comparison results of the 3 precipitation products.

| Floods | Type | Average precipitation (mm) | Relative bias % |
|---|---|---|---|
| | rain gauge | 1.37 | |
| 200806090200 | WRF QPF | 1.55 | 13 |
| | PERSIANN-CCS QPEs | 1.22 | -11 |
| | rain gauge | 0.74 | |
| 200906090800 | WRF QPF | 0.88 | 19 |
| | PERSIANN-CCS QPEs | 0.62 | -16 |
| | rain gauge | 0.42 | |
| 201106010900 | WRF QPF | 0.46 | 10 |
| | PERSIANN-CCS QPEs | 0.39 | -7 |
| | rain gauge | 0.78 | |
| 201206022000 | WRF QPF | 0.95 | 22 |
| | PERSIANN-CCS QPEs | 0.63 | -19 |
| | rain gauge | 0.53 | |
| 201306011400 | WRF QPF | 0.65 | 23 |
| | PERSIANN-CCS QPEs | 0.43 | -20 |
| | rain gauge | 0.77 | |
| average value | WRF QPF | 0.90 | 17 |
| | PERSIANN-CCS QPEs | 0.66 | -14 |

Table 2. Evaluation indices for the karst floods simulation effects.

| Floods | The Nash–Sutcliffe coefficient/C | The correlation coefficient/R | The process relative error/P% | The peak flow relative error/E% | The coefficient of water balance/W | The peak time error/T(hour) |
|---|---|---|---|---|---|---|
| 2005061400 | 0.87 | 0.92 | 0.2 | 0.13 | 1.08 | -7 |
| 2006060400 | 0.91 | 0.89 | 0.17 | 0.07 | 0.92 | -5 |
| 2007070800 | 0.89 | 0.93 | 0.14 | 0.09 | 1.12 | -8 |
| 2008060900 | 0.93 | 0.95 | 0.08 | 0.05 | 0.94 | -3 |

Table 3. The calculation results of the coupled model parameters sensitivity.

| Floods | Potential evaporation/$E_p$ | Evaporation coefficient/$\lambda$ | Wilting percentage/$C_{wl}$ | The saturated water content/$\theta_{sat}$ | The saturation permeability coefficient/$\theta_s$ | The macro crack volume ratio/$V$ |
|---|---|---|---|---|---|---|
| | 0.05 | 0.06 | 0.04 | 0.9 | 0.88 | 0.75 |
| | The field capacity/$\theta_{fc}$ | The soil layer thickness/$z$ | The saturated hydraulic conductivity/$K_s$ | The soil coefficient/$b$ | The bottom slope/$S_p$ | The bottom width/$S_w$ |
| 2005061400 | 0.86 | 0.67 | 0.83 | 0.66 | 0.36 | 0.48 |
| | The slope roughness/$n$ | The channel roughness/$n_1$ | The depletion coefficient /$\omega$ | The permeability coefficient /$K$ | The specific yield of the aquifer /$\chi$ | Thickness of the karst aquifer/$h$ |


|  |  |  |  |  |  |  |
| --- | --- | --- | --- | --- | --- | --- |
|  | 0.25 | 0.17 | 0.13 | 0.75 | 0.73 | 0.68 |
|  | Potential evaporation/$E_p$ | Evaporation coefficient/$\lambda$ | Wilting percentage/$C_{wl}$ | The saturated water content/$\theta_{sat}$ | The saturation permeability coefficient/$\theta_s$ | The macro crack volume ratio/$V$ |
|  | 0.07 | 0.13 | 0.05 | 0.95 | 0.91 | 0.83 |
| 2006060400 | The field capacity/$\theta_{fc}$ | The soil layer thickness/$z$ | The saturated hydraulic conductivity/$K_s$ | The soil coefficient/$b$ | The bottom slope/$S_p$ | The bottom width/$S_w$ |
|  | 0.9 | 0.64 | 0.89 | 0.6 | 0.55 | 0.59 |
|  | The slope roughness/$n$ | The channel roughness/$n_1$ | The depletion coefficient /$\omega$ | The permeability coefficient /$K$ | The specific yield of the aquifer /$\chi$ | Thickness of the karst aquifer/$h$ |
|  | 0.3 | 0.27 | 0.14 | 0.75 | 0.73 | 0.69 |
|  | Potential evaporation/$E_p$ | Evaporation coefficient/$\lambda$ | Wilting percentage/$C_{wl}$ | The saturated water content/$\theta_{sat}$ | The saturation permeability coefficient/$\theta_s$ | The macro crack volume ratio/$V$ |
|  | 0.14 | 0.24 | 0.08 | 0.92 | 0.84 | 0.75 |
| 2007070800 | The field capacity/$\theta_{fc}$ | The soil layer thickness/$z$ | The saturated hydraulic conductivity/$K_s$ | The soil coefficient/$b$ | The bottom slope/$S_p$ | The bottom width/$S_w$ |
|  | 0.81 | 0.63 | 0.77 | 0.61 | 0.51 | 0.57 |
|  | The slope roughness/$n$ | The channel roughness/$n_1$ | The depletion coefficient /$\omega$ | The permeability coefficient /$K$ | The specific yield of the aquifer /$\chi$ | Thickness of the karst aquifer/$h$ |
|  | 0.45 | 0.4 | 0.31 | 0.7 | 0.69 | 0.68 |
|  | Potential evaporation/$E_p$ | Evaporation coefficient/$\lambda$ | Wilting percentage/$C_{wl}$ | The saturated water content/$\theta_{sat}$ | The saturation permeability coefficient/$\theta_s$ | The macro crack volume ratio/$V$ |
|  | 0.18 | 0.26 | 0.11 | 0.94 | 0.92 | 0.78 |
| 2008060900 | The field capacity/$\theta_{fc}$ | The soil layer thickness/$z$ | The saturated hydraulic conductivity/$K_s$ | The soil coefficient/$b$ | The bottom slope/$S_p$ | The bottom width/$S_w$ |
|  | 0.88 | 0.73 | 0.82 | 0.64 | 0.53 | 0.6 |
|  | The slope roughness/$n$ | The channel roughness/$n_1$ | The depletion coefficient /$\omega$ | The permeability coefficient /$K$ | The specific yield of the aquifer /$\chi$ | Thickness of the karst aquifer/$h$ |
|  | 0.47 | 0.45 | 0.36 | 0.8 | 0.75 | 0.72 |








952 Table 4. The evaluation indices of karst floods simulations with the original WRF QPF and
953 PERSIANN-CCS QPEs and their postprocessed values.

| Floods | Types | The Nash–Sutcliffe coefficient/C | The correlation coefficient/R | The process relative error/P% | The peak flow relative error/E% | The coefficient of water balance/W | The peak time error/T(h) |
|---|---|---|---|---|---|---|---|
| 200806090000 | WRF QPF | 0.72 | 0.80 | 25 | 18 | 1.02 | -9 |
| | The postprocessed WRF QPF | 0.78 | 0.82 | 20 | 13 | 0.95 | -7 |
| | PERSIANN-CCS QPEs | 0.76 | 0.83 | 21 | 6 | 0.92 | -10 |
| | The postprocessed PERSIANN-CCS QPEs | 0.83 | 0.88 | 18 | 5 | 0.94 | -4 |
| 200906090800 | WRF QPF | 0.81 | 0.82 | 24 | 20 | 1.12 | -6 |
| | The postprocessed WRF QPF | 0.83 | 0.83 | 20 | 14 | 1.06 | -4 |
| | PERSIANN-CCS QPEs | 0.82 | 0.81 | 28 | 18 | 0.79 | -6 |
| | the postprocessed PERSIANN-CCS QPEs | 0.85 | 0.87 | 22 | 12 | 0.85 | -3 |
| 201106010900 | WRF QPF | 0.79 | 0.81 | 26 | 14 | 1.15 | -7 |
| | The postprocessed WRF QPF | 0.83 | 0.83 | 20 | 10 | 1.08 | -6 |
| | PERSIANN-CCS QPEs | 0.85 | 0.85 | 21 | 12 | 0.92 | -8 |
| | The postprocessed PERSIANN-CCS QPEs | 0.91 | 0.87 | 19 | 6 | 0.94 | -6 |
| 20120602200 | WRF QPF | 0.78 | 0.82 | 18 | 13 | 1.28 | -10 |
| | The postprocessed WRF QPF | 0.81 | 0.83 | 10 | 11 | 1.15 | -8 |
| | PERSIANN-CCS QPEs | 0.86 | 0.84 | 16 | 15 | 0.78 | -7 |
| | the postprocessed PERSIANN-CCS QPEs | 0.92 | 0.89 | 9 | 6 | 0.85 | -4 |
| 201306011400 | WRF QPF | 0.78 | 0.82 | 13 | 21 | 1.20 | -8 |
| | The postprocessed WRF QPF | 0.82 | 0.85 | 9 | 12 | 1.12 | -6 |





| Floods | Type | Nash–Sutcliffe coefficient/C | Correlation coefficient/R | Process relative error/P% | Peak flow relative error/E% | The coefficient of water balance/W | Peak time error/T (hour) |
|---|---|---|---|---|---|---|---|
| | PERSIANN-CCS QPEs | 0.82 | 0.89 | 12 | 17 | 0.85 | -5 |
| | The postprocessed PERSIANN-CCS QPEs | 0.86 | 0.91 | 8 | 9 | 0.87 | -4 |
| | WRF QPF | 0.78 | 0.81 | 21 | 17 | 1.15 | -8 |
| | The postprocessed WRF QPF | 0.81 | 0.83 | 16 | 12 | 1.07 | -6 |
| average value | PERSIANN-CCS QPEs | 0.82 | 0.84 | 20 | 14 | 0.85 | -7 |
| | The postprocessed PERSIANN-CCS QPEs | 0.87 | 0.88 | 15 | 8 | 0.89 | -4 |

Table 5. The evaluation indices of karst floods simulations with the 3 precipitation products.

| Floods | Type | Nash–Sutcliffe coefficient/C | Correlation coefficient/R | Process relative error/P% | Peak flow relative error/E% | The coefficient of water balance/W | Peak time error/T (hour) |
|---|---|---|---|---|---|---|---|
| 200806090000 | rain gauge | 0.85 | 0.91 | 15 | 3 | 0.89 | -6 |
| | WRF QPF | 0.78 | 0.82 | 20 | 13 | 0.95 | -7 |
| | PERSIANN-CCS QPEs | 0.83 | 0.88 | 18 | 5 | 0.94 | -4 |
| 200906090800 | rain gauge | 0.95 | 0.92 | 17 | 4 | 0.9 | -2 |
| | WRF QPF | 0.83 | 0.83 | 20 | 14 | 1.06 | -4 |
| | PERSIANN-CCS QPEs | 0.85 | 0.87 | 22 | 12 | 0.85 | -3 |
| 201106010900 | rain gauge | 0.95 | 0.92 | 16 | 3 | 1.02 | -7 |
| | WRF QPF | 0.83 | 0.83 | 20 | 10 | 1.08 | -6 |
| | PERSIANN-CCS QPEs | 0.91 | 0.87 | 19 | 6 | 0.94 | -6 |
| 20120602200 | rain gauge | 0.93 | 0.91 | 8 | 5 | 0.89 | -6 |
| | WRF QPF | 0.81 | 0.83 | 10 | 11 | 1.15 | -8 |
| | PERSIANN-CCS QPEs | 0.92 | 0.89 | 9 | 6 | 0.85 | -4 |
| 201306011400 | rain gauge | 0.95 | 0.94 | 7 | 6 | 0.92 | -4 |
| | WRF QPF | 0.82 | 0.85 | 9 | 12 | 1.12 | -6 |
| | PERSIANN-CCS QPEs | 0.86 | 0.91 | 8 | 9 | 0.87 | -4 |
| average value | rain gauge | 0.93 | 0.92 | 13 | 4 | 0.92 | -5 |
| | WRF QPF | 0.81 | 0.83 | 16 | 12 | 1.07 | -6 |
| | PERSIANN-CCS QPEs | 0.87 | 0.88 | 15 | 8 | 0.89 | -4 |




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
