# Peer review of "Karst-Liuxihe model"

_Hydrology and Earth System Sciences, 2019_

## Referee Comment (RC1) · Anonymous Referee #1 · 3 Sep 2019

In this submitted manuscript, authors (1) evaluated the QPEs from PERSIANN-CCS product, and the WRF QPF developed for Karst regions, (2) developed a karst region-specific hydrologic model, termed Karst-Liuxihe Model by adding some enhancement modules to an existing Liuxihe hydrologic model. The current manuscript suffers from several major issues and the referee cannot suggest acceptance.

Major issues: 1. Reviewer is not convinced about how the proposed Karst-Liuxihe model could address the challenges of hydrologic simulation over Karst areas. Authors mentioned two shortcomings of using distributed hydrologic models in lines 156-167, with one challenge being lack of in-situ data, and another of high computational efficiency of parameter calibration. Authors also summarized the way this study tries to address these two challenges by: (1) doing "field survey and tracing test" and "collect data from internet at no cost", and (2) using "An improved Particle Swarm Optimization method (Chen et al. 2016)" for parameter estimation, respectively. However, neither of these approaches are based on the innovation of modeling scheme, rather, they are commonly standards for obtaining data or calibrating hydrologic models. Therefore, the novelty and contributions of this study are questionable.

2. The organization of the entire introduction is very confusing. Sometimes, authors talked about features of Karst regions, remote sensing v.s. in-situ data, forecast models, distributed hydrologic models vs lump hydrologic models, data needs and challenges, development of the proposed Karst-liuxihe model, the lead time and resolutions, and tangibly relevant literature without any detailed summary of their experiments and conclusions. As a result, reviewer is not able to identify the following (1) background and Motivation of this study, (2) novelty and contributions, (3) methodology developments, (4) advantages of the proposed methodology, and (5) how authors demonstrate the hypothesis or conclusion by designed studies.

3. With respect to the results of Figure 3-7, which show the comparison among WPF QPF, PERSIANN-CCS QPE and gauge observation, Reviewer has two major concerns. (1) how fair it is to compare a precipitation forecasting product with an estimation product? It becomes the essential issue of comparing "apple" to "orange". Comparison can be done, however, any results or conclusions drawn from such type of comparison are based on the assumptions that "apple" and "orange" have the same mechanisms, physical dynamics, and functions. Unfortunately, they are not. QPE products shall be compared with gauge or other QPE products, such as comparing different remote sensing products over the same region. QPF products shall only be compared to ground truth or other QPF products with the same lead time. (2) A second concern is that authors have claimed several times that the reliability of gauge precipitation and the lack of data are the main motivations of this study to use remote sensing and model

forecasts data to drive their proposed hydrologic models. Now with the direct comparison with gauge observations, the gauge precipitation is used as a reference to evaluate two precipitation products, in which the underline assumption becomes that the gauge precipitation is true and accurate in the Karst areas. If this is the case, why bother using another remote sensing product to estimate precipitation over the Karst areas? Since both gauge and PERSIANN-CCS are historical or near-real-time observations. This underline assumption, though not directly mentioned, contradicts the design of experiments and motivations of using WRP QPF and PERSIANN CCS QPE as inputs to their hydrologic simulations.

4. Similar to previous concern, the post-processing steps to obtain gauge corrected average PERSIANN-CCS and WRF-QPF in equation 1-3, as well as section 3.4 (post-processing of the 2 weather models) rely on the same assumption that gauge networks are more reliable than others over the study region. Then, why bother to use another precipitation estimation product without any lead-time?

5. There are a quite amount of results presented by authors on 5 different floods simulations, using the original and bias-corrected WRF QPF and PERSIANN-CCS. It is not surprising that the bias-corrected (post-processed) WRF QPF and PERSIANN-CCS have better statistics than the simulations than the original inputs. The main concern reviewer has is that what evidence proves that the proposed Karst-Liuxihe Model is better than the original version of Liuxihe model? It seemed all simulations are with the proposed model without any baseline comparison to its original version, and repeated but same simulation do not necessarily add values to prove the advantages of the proposed model.

6. Authors first conclusion is that "The postprocessing method proposed in this study could largely reduce these relative errors." However, reviewer finds in Table 4 that even the original data was not corrected, it still leads to a relative good simulation of floods with NASH values about 0.7 or above. If this is the case, why we need to postprocessing the WRF-QPF and PERSIANN-CCS? We can still obtain good flood simulation by

tuning the hydrologic model parameters.

7. Authors conclusion No.4 on "The flood processes simulated by the Karst-Liuxihe model using the rain gauge precipitation were the best". The use of gauge should be a baseline and not a conclusion. The model is calibrated using the gauge, and of course, the rain gauge precipitation could produce the best flood simulation.

8. The authors include the lead time error of both WRF-QPF and PERSIANN-CCS in Tables 4 and 5. All simulations have negative peak time error, T, also see authors last conclusion. Reviewer is wondering how to interpret this number. Do negative values mean the peak is predicted to happen prior to the actual flood peak? or the other way? If it means a delay in simulation, how we use these data to timely predict potential floods? If this is prior to the flood peak, reviewer is wondering how PERSIANN-CCS and gauge inputs can produce a lead time given the data itself is historical precipitation estimates instead of forecasts into the future.

9. Authors seem to know but are reluctant to provide more background information on using QPF or QPE on hydrologic simulations over Karst areas. Line 77-78 says that "only a few studies of rainfall forecasting based on WRF QPF and PERSIANN-CCS QPEs have been conducted in karst areas until now, and even if there are studies, the practical accuracy is generally poor." Are there any relevant studies? Reviewer believes there should be lots. What will be the differences between this one and other literature?

10. Another concern is the investigation of parameter sensitivity. In this study, authors applied NSE to draw the conclusion about parameter sensitivity (Line 654-656 and Table 3) without perturbing each individual parameter. The current use of NSE only demonstrates how well the simulation can represent the observation. But, it has nothing to do with the sensitivity of parameter in the hydrologic model. More explanations of how authors evaluate the parameter sensitivity are needed.

Minor issues:

Abstract Line 20-27: This sentence runs for more than 8 lines without any clear structure. Please break into 2-3 smaller sentences for better readability.

Line 66-68: this sentence is very cumbersome, The parts of "enable . . .. To be easily obtained" needs to be rewritten.

Line 84-86: there are a few grammar errors in this sentence. Please re-write.

Line 104-106: Ground gauge, of course, has no lead time as compared to the forecasting model. Not sure why authors emphasize on this known fact.

Line 108: grammar error. "People" cannot be "transferred". Do the authors mean "evacuate"?

Line 209: "The channel length of Liujiang river is about 1120 km and the area is about $5.8 \times 104$ km2". What area is about $5.8 \times 104$ km2 here? The area of water surface of the river?

Line 265&267: Why use the phrase "property data" here? What is the meaning of it?

Line 350-351: The reviewer believes the authors are trying to say "the rainfall distributions of WRF QPF, the PERSIANN-CCS, and the observed Precipitation data appears to be quite similar to each other".

Line 385, and Line 399-404: In equation (1), the reviewer presumes the unit of $P_i$ should be mm/cm/m, and the unit of $F_i$ should be a mm2/cm2/m2. Thus, $P_i$ times $F_i$ should give us the volume of water here. Then $F_i \times P_i / N$ is still in volume. Later postprocessing procedure cannot be continued according to the instruction. It is important to give correct and detailed steps and evaluation of the applied postprocessing of two precipitation inputs.

Line 329-344 It seems like the authors did not evaluate precipitation product for the year of 2010. Is it because there was no flood event occurred in 2010? The following question would be why evaluate two precipitation product's performance only under

several flood periods? Why not evaluate both precipitation products over as a long time period as possible?

Line 473: It might be better to use "improve" instead of "increase" in the section title.

Line 646 It might be better to say "the MPSA was modified and improved from the GLUE algorithm".

Line 842: It is very difficult to tell the difference between the three types of rain gauges on Fig1a. Suggesting change high contrast color combination.

The abstract seems to be very long without a concise focus on the scope of work and summary of novelty and motivation of this study. It reads like an introduction instead of a concise abstract.

Unsupported claim on "Among these weather models, WRF QPF and PERSIANN-CCS QPEs may be better ways to acquire precipitation results effectively in karst basins." Why WRF QPF and PERSIANN-CCS are selected for this study? References are needed.

---

## Author Comment (AC1) · 16 Sep 2019

**Reply to the comments of Anonymous Referee #1-AC1**

Anonymous Referee #1

In this submitted manuscript, authors (1) evaluated the QPEs from PERSIANN-CCS product, and the WRF QPF developed for Karst regions, (2) developed a karst regionspecific hydrologic model, termed Karst-Liuxihe Model by adding some enhancement modules to an existing Liuxihe hydrologic model. The current manuscript suffers from several major issues and the referee cannot suggest acceptance.

Major issues: 1. Reviewer is not convinced about how the proposed Karst-Liuxihe model could address the challenges of hydrologic simulation over Karst areas. Authors mentioned two shortcomings of using distributed hydrologic models in lines 156-167, with one challenge being lack of in-situ data, and another of high computational efficiency of parameter calibration. Authors also summarized the way this study tries to address these two challenges by: (1) doing "field survey and tracing test" and "collect data from internet at no cost", and (2) using "An improved Particle Swarm Optimization method (Chen et al. 2016)" for parameter estimation, respectively. However, neither of these approaches are based on the innovation of modeling scheme, rather, they are commonly standards for obtaining data or calibrating hydrologic models. Therefore, the novelty and contributions of this study are questionable.

2. The organization of the entire introduction is very confusing. Sometimes, authors talked about features of Karst regions, remote sensing v.s. in-situ data, forecast models, distributed hydrologic models vs lump hydrologic models, data needs and challenges, development of the proposed Karst-liuxihe model, the lead time and resolutions, and tangibly relevant literature without any detailed summary of their experiments and conclusions. As a result, reviewer is not able to identify the following (1) background and Motivation of this study, (2) novelty and contributions, (3) methodology developments, (4) advantages of the proposed ethodology, and (5) how authors demonstrate the hypothesis or conclusion by designed studies.

3. With respect to the results of Figure 3-7, which show the comparison among WPF QPF, PERSIANN-CCS QPE and gauge observation, Reviewer has two major concerns.

(1) how fair it is to compare a precipitation forecasting product with an estimation product? It becomes the essential issue of comparing "apple" to "orange". Comparison can be done, however, any results or conclusions drawn from such type of comparison are based on the assumptions that "apple" and "orange" have the same mechanisms, physical dynamics, and functions. Unfortunately, they are not. QPE products shall be compared with gauge or other QPE products, such as comparing different remote sensing products over the same region. QPF products shall only be compared to ground truth or other QPF products with the same lead time.

(2) A second concern is that authors have claimed several times that the reliability of gauge precipitation and the lack of data are the main motivations of this study to use remote sensing and model forecasts data to drive their proposed hydrologic models. Now with the direct comparison with gauge observations, the gauge precipitation is used as a reference to evaluate two precipitation products, in which the underline assumption becomes that the gauge

precipitation is true and accurate in the Karst areas. If this is the case, why bother using another remote sensing product to estimate precipitation over the Karst areas?

Since both gauge and PERSIANN-CCS are historical or near-real-time observations. This underline assumption, though not directly mentioned, contradicts the design of experiments and motivations of using WRP QPF and PERSIANN CCS QPE as inputs to their hydrologic simulations.

4. Similar to previous concern, the post-processing steps to obtain gauge corrected average PERSIANN-CCS and WRF-QPF in equation 1-3, as well as section 3.4 (postprocessing of the 2 weather models) rely on the same assumption that gauge networks are more reliable than others over the study region. Then, why bother to use another precipitation estimation product without any lead-time?

5. There are a quite amount of results presented by authors on 5 different floods simulations, using the original and bias-corrected WRF QPF and PERSIANN-CCS. It is not surprising that the bias-corrected (post-processed) WRF QPF and PERSIANNCCS have better statistics than the simulations than the original inputs. The main concern reviewer has is that what evidence proves that the proposed Karst-Liuxihe Model is better than the original version of Liuxihe model? It seemed all simulations are with the proposed model without any baseline comparison to its original version, and repeated but same simulation do not necessarily add values to prove the advantages of the proposed model.

6. Authors first conclusion is that "The postprocessing method proposed in this study could largely reduce these relative errors." However, reviewer finds in Table 4 that even the original data was not corrected, it still leads to a relative good simulation of floods with NASH values about 0.7 or above. If this is the case, why we need to postprocessing the WRF-QPF and PERSIANN-CCS? We can still obtain good flood simulation by tuning the hydrologic model parameters.

7. Authors conclusion No.4 on "The flood processes simulated by the Karst-Liuxihe model using the rain gauge precipitation were the best". The use of gauge should be a baseline and not a conclusion. The model is calibrated using the gauge, and of course, the rain gauge precipitation could produce the best flood simulation.

8. The authors include the lead time error of both WRF-QPF and PERSIANN-CCS in Tables 4 and 5. All simulations have negative peak time error, T, also see authors last conclusion. Reviewer is wondering how to interpret this number. Do negative values mean the peak is predicted to happen prior to the actual flood peak? or the other way? If it means a delay in simulation, how we use these data to timely predict potential floods? If this is prior to the flood peak, reviewer is wondering how PERSIANN-CCS and gauge inputs can produce a lead time given the data itself is historical precipitation estimates instead of forecasts into the future.

9. Authors seem to know but are reluctant to provide more background information on using QPF or QPE on hydrologic simulations over Karst areas. Line 77-78 says that "only a few studies of rainfall forecasting based on WRF QPF and PERSIANN-CCS QPEs have been conducted in karst areas until now, and even if there are studies, the practical accuracy is generally poor." Are there any relevant studies? Reviewer believes there should be lots. What will be the differences between this one and other literature?

10. Another concern is the investigation of parameter sensitivity. In this study, authors

applied NSE to draw the conclusion about parameter sensitivity (Line 654-656 and Table 3) without perturbing each individual parameter. The current use of NSE only demonstrates how well the simulation can represent the observation. But, it has nothing to do with the sensitivity of parameter in the hydrologic model. More explanations of how authors evaluate the parameter sensitivity are needed.

Minor issues:

Abstract Line 20-27: This sentence runs for more than 8 lines without any clear structure. Please break into 2-3 smaller sentences for better readability.

Line 66-68: this sentence is very cumbersome, The parts of "enable ... To be easily obtained" needs to be rewritten.

Line 84-86: there are a few grammar errors in this sentence. Please re-write.

Line 104-106: Ground gauge, of course, has no lead time as compared to the forecasting model. Not sure why authors emphasize on this known fact.

Line 108: grammar error. "People" cannot be "transferred". Do the authors mean "evacuate"?

Line 209: "The channel length of Liujiang river is about 1120 km and the area is about 5.8_104 km2". What area is about 5.8_104 km2 here? The area of water surface of the river?

Line 265&267: Why use the phrase "property data" here? What is the meaning of it?

Line 350-351: The reviewer believes the authors are trying to say "the rainfall distributions of WRF QPF, the PERSIANN-CCS, and the observed Precipitation data appears to be quite similar to each other".

Line 385, and Line 399-404: In equation (1), the reviewer presumes the unit of Pi should be mm/cm/m, and the unit of Fi should be a mm2/cm2/m2. Thus, Pi times Fi should give us the volume of water here. Then Fi_Pi / N is still in volume. Later postprocessing procedure cannot be continued according to the instruction. It is important to give correct and detailed steps and evaluation of the applied postprocessing of two precipitation inputs.

Line 329-344 It seems like the authors did not evaluate precipitation product for the year of 2010. Is it because there was no flood event occurred in 2010? The following question would be why evaluate two precipitation product's performance only under several flood periods? Why not evaluate both precipitation products over as a long time period as possible?

Line 473: It might be better to use "improve" instead of "increase" in the section title.

Line 646 It might be better to say "the MPSA was modified and improved from the GLUE algorithm".

Line 842: It is very difficult to tell the difference between the three types of rain gauges on Fig1a. Suggesting change high contrast color combination.

The abstract seems to be very long without a concise focus on the scope of work and summary of novelty and motivation of this study. It reads like an introduction instead of a concise abstract.

Unsupported claim on "Among these weather models, WRF QPF and PERSIANN-CCS QPEs may be better ways to acquire precipitation results effectively in karst basins."
Why WRF QPF and PERSIANN-CCS are selected for this study? References are needed.

We thank the referee very much for reviewing the manuscript. The following are our point-by-point responses to the reviewer's comments.

Comment 1.

Major issues: 1. Reviewer is not convinced about how the proposed Karst-Liuxihe model could address the challenges of hydrologic simulation over Karst areas. Authors mentioned two shortcomings of using distributed hydrologic models in lines 156-167, with one challenge being lack of in-situ data, and another of high computational efficiency of parameter calibration. Authors also summarized the way this study tries to address these two challenges by: (1) doing "field survey and tracing test" and "collect data from internet at no cost", and (2) using "An improved Particle Swarm Optimization method (Chen et al. 2016)" for parameter estimation, respectively. However, neither of these approaches are based on the innovation of modeling scheme, rather, they are commonly standards for obtaining data or calibrating hydrologic models. Therefore, the novelty and contributions of this study are questionable.

Response:

There are two limitations of using distributed hydrologic models in Karst areas. One challenge is the lack of in-situ data; such data are needed to build a distributed hydrological model. These data include hydrogeological data and meteorological data. Regarding the lack of hydrogeological data, data such as high-resolution DEM, land use and soil type data can be downloaded from the internet at no cost or collected via field surveys or tracing tests. Indeed, as the reviewer said, these are commonly used approaches for obtaining data or calibrating hydrologic models. However, in karst areas, the common problem of the lack of meteorological data cannot be solved by performing field surveys and tracing tests or compiling data from the internet. Because of the complex topography of karst areas, most of which are occupied by steep mountains, few rainfall stations have been built in karst regions. This makes it difficult to obtain reliable, long-term rainfall data. Some rainfall data were obtained from rain gauges in this study, which were used to correct the rainfall results of the two meteorological models. However, within the entire research area ($5.8 \times 10^4$ km$^2$), there are only 66 rain gauges, with nearly 900 square kilometres per rain gauge. The representativeness of rainfall results obtained from rain gauges is seriously insufficient. Therefore, this study established two meteorological models (WRF QPF and PERSIANN-CCS QPEs) to obtain reliable rainfall data for karst areas. Although some studies have applied WRF QPF to karst basins, but the lead times of these WRF QPF models are short, usually only 24 hours. Such a model with a shorter prediction period can forecast rainfall products with higher accuracy. In the present study, the lead time of the WRF QPF model can reach 96 hours. This lead time guarantees the accuracy of rainfall forecast. This provides a long response time for flood forecasting by coupling the WRF QPF model with the Karst-Liuxihe Model. There are many studies on the PERSIANN system and its subsystem PERSIANN-CDR (Climate Data Record), which have achieved good results. However, there are only a few studies of PERSIANN-CCS (Cloud Classification System) in karst areas. These analyses using WRF and PERSIANN models will be added to the revised version, and the advantages of the two meteorological models in this paper are compared with previous models will be discussed in the revised version.

The other limitation of distributed hydrological models in karst areas is the problem of model calculation efficiency. The reviewer refers to "An improved Particle Swarm Optimization method (Chen et al. 2016")" used for parameter estimation in the manuscript and states that this approach is not an innovation of the present study. A contribution of this paper is that the uncertainty in this improved Particle Swarm Optimization method is studied. Specific and detailed parametric uncertainty calculations will be added to the revised version. In the study by Chen et al. (2016), there is no parametric uncertainty analysis.

The most important innovation of the present research lies in the improvement and perfection of the Liuxihe model structure and function. A new hydrologic model, i.e., the Karst-Liuxihe Model, is proposed, which has some enhancement modules added; this Karst-Liuxihe Model has never appeared in the previous literature. The improvement of Liuxihe model is extensive and includes the division of the model into the smallest structural units, termed karst hydrology response units (KHRUs); and new algorithms of rainfall-runoff, especially for the confluence of karst groundwater. The original Liuxihe Model can only be used to simulate the surface river and cannot be used reliably in karst areas. The Karst-Liuxihe Model does well in karst areas and is effective for calculating the exchange of water between surface water and karst underground rivers.

Comment 2.
The organization of the entire introduction is very confusing. Sometimes, authors talked about features of Karst regions, remote sensing v.s. in-situ data, forecast models, distributed hydrologic models vs lump hydrologic models, data needs and challenges, development of the proposed Karst-liuxihe model, the lead time and resolutions, and tangibly relevant literature without any detailed summary of their experiments and conclusions. As a result, reviewer is not able to identify the following (1) background and Motivation of this study, (2) novelty and contributions, (3) methodology developments, (4) advantages of the proposed ethodology, and (5) how authors demonstrate the hypothesis or conclusion by designed studies.

Response:

The introduction will be restructured for clarity in the revised version . Some details of our field experiments and conclusions will be added . In addition, in the revised version of the paper, additional literature about previous models including the WRF, PERSIANN models as well as the original Liuxihe model and other distributed hydrological models will be cited, and the improvements of our model over previous models will be discussed. We will revise the introduction section to describe the research purpose and innovation of this study, the developments and advantages of the model method and the rationale for this research.

Comment 3.
With respect to the results of Figure 3-7, which show the comparison among WPF QPF, PERSIANN-CCS QPE and gauge observation, Reviewer has two major concerns.

(1) how fair it is to compare a precipitation forecasting product with an estimation product? It becomes the essential issue of comparing "apple" to "orange". Comparison can be done, however, any results or conclusions drawn from such type of comparison are based on the assumptions that "apple" and "orange" have the same mechanisms, physical dynamics, and functions. Unfortunately, they are not. QPE products shall be compared with gauge or other QPE products, such as comparing different remote sensing products over the same region. QPF products shall only be compared to ground truth or other QPF products with the same lead time.

(2) A second concern is that authors have claimed several times that the reliability of gauge precipitation and the lack of data are the main motivations of this study to use remote sensing and model forecasts data to drive their proposed hydrologic models. Now with the direct comparison with gauge observations, the gauge precipitation is used as a reference to evaluate two precipitation products, in which the underline assumption becomes that the gauge precipitation is true and accurate in the Karst areas. If this is the case, why bother using another remote sensing product to estimate precipitation over the Karst areas?

Since both gauge and PERSIANN-CCS are historical or near-real-time observations. This underline assumption, though not directly mentioned, contradicts the design of experiments and motivations of using WRP QPF and PERSIANN CCS QPE as inputs to their hydrologic simulations.

Response:

Figures 3-7 show the comparisons among WPF QPF, PERSIANN-CCS QPE and the gauge observations. The reviewer points out (concern 1) that it is perhaps inappropriate to compare a precipitation-forecasting product, i.e., WPF QPF, with an estimation product, i.e., PERSIANN-CCS QPE. These rainfall products have two different mechanisms. The purpose of this paper is not to compare the characteristics of these two meteorological products in terms of mechanism. Rather, the purpose is to identify a meteorological product suitable for karst areas, regardless of whether it is a QPF, QPE or other weather models, such as weather radar, because rainfall gauges are scarce in karst areas and cannot provide reliable rainfall data. There are several differences between WPF QPF and PERSIANN-CCS QPE; for example, the rainfall forecast results by WPF QPF have a lead time of 96 hours, whereas the PERSIANN-CCS QPE has no lead time. The title of the manuscript is "Comparing the performances of WRF QPF and PERSIANN-CCS QPEs in karst flood **simulations** and **forecasting** with a new Karst-Liuxihe model". The purpose of coupling WRF QPF with the Karst-Liuxihe model in this paper is to **forecast** floods, which is possible due to the long lead time offered by the WRF QPF. In contrast, the purpose coupling PERSIANN-CCS QPEs with the hydrologic model is to **simulate** and perform inverse analysis of karst floods.

The reviewer want to know (concern 2) why the second remote sensing product, i.e., PERSIANN CCS QPE, was used in this study, since both gauge and PERSIANN-CCS data represent historical or near-real-time observations. The PERSIANN-CCS product and rain gauges are similar in that there is no lead time in the mechanisms. However, in terms of rainfall volume, the rainfall observed by rain gauge can indicate the actual rainfall in the basin. In contrast, the rainfall estimated by PERSIANN-CCS has errors and cannot be regarded as the real rainfall in the basin. However, as a weather model, PERSIANN-CCS can provide effective rainfall estimation results in the absence of rain gauges in karst watersheds.

The purposes of using these two weather models to obtain rainfall in the present study are different. As mentioned above, the purpose of coupling WRF QPF with the hydrologic model in this paper is to forecast karst floods, whereas the purpose of coupling PERSIANN-CCS QPEs with the hydrologic model is to perform flood simulation.

Comment 4.

Similar to previous concern, the post-processing steps to obtain gauge corrected average PERSIANN-CCS and WRF-QPF in equation 1-3, as well as section 3.4 (postprocessing of the 2 weather models) rely on the same assumption that gauge networks are more reliable than others over the study region. Then, why bother to use another precipitation estimation product without any lead-time?

Response:

As mentioned above, the rainfall observations by rain gauge are not fully representative of actual rainfall in the study area. Therefore, this study established two meteorological models (WRF QPF and PERSIANN-CCS QPEs) to obtain reliable rainfall data in karst areas. The reviewer want to know why PERSIANN CCS QPE rainfall data are used in this study given that WRF QPF was used to forecast rainfall in the basin and that both rain gauge and PERSIANN-CCS data are historical and have no lead-time. As previously mentioned, unlike the rainfall observed by rain gauge, the rainfall estimated by PERSIANN-CCS has errors regarding rainfall magnitude and cannot be regarded as the real rainfall in the basin. However, if there are no rain gauges in the study area (and the lack of rain gauge is a common problem in karst areas), the rainfall estimated by PERSIANN-CCS can be used to approximate actual rainfall in the basin and to correct the forecast results of other weather models, such as WRF QPF. This means that the results of this study can be generalized to other karst areas without rain gauges. If forecasting of karst floods is desired, the rainfall in the study area must be forecast first, and then WRF QPF can be adopted. PERSIANN-CCS is also indispensable and can be used to correct the rainfall forecast results of the WRF model. Of course, considering that the rainfall results estimated by PERSIANN-CCS also have some errors, in fact, this paper finds that there is indeed such an error by comparing it with the results of rain gauge. The average relative error was -14% between the PERSIANN-CCS QPEs and precipitation by rain gauge. Previous research by the authors (Li et al., 2019) also found that the rainfall results estimated by PERSIANN-CCS are smaller than that observed rainfall by rain gauge. Therefore, this underestimate problem should be taken into account when using the PERSIANN-CCS QPEs to correct other weather models like the WRF QPF.

Comment 5.

There are a quite amount of results presented by authors on 5 different floods simulations, using the original and bias-corrected WRF QPF and PERSIANN-CCS. It is not surprising that the bias-corrected (post-processed) WRF QPF and PERSIANNCCS have better statistics than the simulations than the original inputs. The main concern reviewer has is that what evidence proves that the proposed Karst-Liuxihe Model is better than the original version of Liuxihe model? It seemed all simulations are with the proposed model without any baseline comparison to its original version, and repeated but same simulation do not necessarily add values to prove the advantages of the proposed model.

Response:

The main concern of the reviewer is that it is unclear what evidence there is proving that the proposed Karst-Liuxihe Model is better than the original Liuxihe model. In fact, as a terrestrial hydrological model, the original Liuxihe model does well in surface river simulation and forecasting in non-karst areas, and many valuable research results have been obtained using this model (Chen et al., 2016,2017; Li et al., 2017). However, it is not suitable for karst areas due to its underlying mechanism. The original Liuxihe model treats the entire underground layer as a whole, and the confluence calculation of the underground river is performed using a linear reservoir method. However, karst groundwater is nonlinear and more complicated than that in non-karst areas. The original Liuxihe model is unable to distinguish karst fissure flow from conduit flow and the hydraulic connection between them. Therefore, the Karst-Liuxihe Model is proposed in this study and is developed by adding some enhancement modules to the original version of Liuxihe model.

Since the reviewer expressed concern over the performance of the improved Karst-Liuxihe Model relative to the original model, a comparison of the two models for karst flood process simulation and forecasting will be added to the revised manuscript.

Comment 6.

Authors first conclusion is that "The postprocessing method proposed in this study could largely reduce these relative errors." However, reviewer finds in Table 4 that even the original data was not corrected, it still leads to a relative good simulation of floods with NASH values about 0.7 or above. If this is the case, why we need to postprocessing the WRF-QPF and PERSIANN-CCS? We can still obtain good flood simulation by tuning the hydrologic model parameters.

Response:

The reviewer asks why postprocessing of the WRF-QPF and PERSIANN-CCS data was necessary. Even without correction of the original data, good simulation of floods is obtained, with NASH values about 0.7 or above. Furthermore, we can achieve good flood simulation by tuning the hydrologic model parameters. It is possible to achieve good results by coupling the original 2 weather models with the Karst-Liuxihe model, which shows that these two weather models can be used to obtain reliable rainfall data in karst areas. However, the rainfall results are not without error. Figure 3-7 and Table 1 show the rainfall results obtained from WPF QPF, PERSIANN-CCS QPE and gauge observations; the original 2 weather models yield some errors in the amount of rainfall relative to the data obtained with rain gauges. The relative error of the weather model precipitation data to the rain gauge data was 17% for WRF QPF and -14% for PERSIANN-CCS QPEs. These results indicate that the data need correction.

Compared with the karst flood events simulations based on the original 2 weather models, the floods simulations with the postprocessed WRF QPF and PERSIANN-CCS QPEs were much better. For example, as shown in Table 4, the average water balance coefficient, Nash-Sutcliffe coefficient, and correlation coefficient for the postprocessed WRF QPF were increased by 8%, 3%, and 2%, respectively, whereas the average peak flow relative error, process relative error, and the peak flow time error were decreased by 5%, 5%, and 2 hours, respectively. The reviewer pointed that in Table 4, even the original data by the two weather models was not corrected, it still leads to a relative good simulation of floods with NASH

values about 0.7 or above. However, we should also see that all 6 flood simulation evaluation indicators besides NASH values have been improved after post-treatment, especially the Process relative error/P% forflood simulation error of WRF model has been reduced from 21% to 16% after post-processing, while the Peak flow relative error/E% has been reduced from 17% to 12%. These two simulation indexes are very important in flood simulation and forecasting, which indicates that the post-processing method can greatly improve the accuracy of flood simulation. Similar results were obtained for the postprocessed PERSIANN-CCS QPEs. Therefore, the postprocessing method proposed for the original 2 weather models in this study is effective and necessary to achieve reliable results.

To obtain good flood simulation results, many aspects can be improved; for example, the model parameters can be adjusted, as noted by the reviewer. However, we can improve the accuracy of the model rainfall input data by the postprocessing method proposed in this study, and there is little reason for us not to do so.

Comment 7.

Authors conclusion No.4 on "The flood processes simulated by the Karst-Liuxihe model using the rain gauge precipitation were the best". The use of gauge should be a baseline and not a conclusion. The model is calibrated using the gauge, and of course, the rain gauge precipitation could produce the best flood simulation.

Response:

We agree with the reviewer's comment that the gauge data are used as reference data and are not themselves being evaluated. Conclusion No. 4 will be revised accordingly in the revised manuscript.

Comment 8.

The authors include the lead time error of both WRF-QPF and PERSIANN-CCS in Tables 4 and 5. All simulations have negative peak time error, T, also see authors last conclusion. Reviewer is wondering how to interpret this number. Do negative values mean the peak is predicted to happen prior to the actual flood peak? or the other way? If it means a delay in simulation, how we use these data to timely predict potential floods? If this is prior to the flood peak, reviewer is wondering how PERSIANN-CCS and gauge inputs can produce a lead time given the data itself is historical precipitation estimates instead of forecasts into the future.

Response:

In Tables 4 and 5, all simulations have negative peak time error, T. The negative values for the rain gauges, WRF QPF, and PERSIANN-CCS QPEs are -5, -6, and -4 hours, respectively. The negative values mean the peak is predicted to happen prior to the actual flood peak. In fact, this negative value represents the systematic error of the model—not only the Karst-Liuxihe model, but also the prototype Liuxihe model. This systematic error is also found in the actual application of the prototype Liuxihe model. We found that after improving the model to the Karst-Liuxihe model, this systematic error persists independent of the type of rainfall data are input to the model (rain gauge, PERSIANN-CCS or WRF data). This systematic error of the model is related to the size of the study area and the length of the confluence. According to previous research (Chen et al., 2016, 2017; Li et al., 2017), when

the basin area is no more than 10,000 km$^2$ and the river length is less than 1,000 km, the time difference of the simulation is typically no more than 1 hour. Such a small time difference is acceptable in actual flood forecasting. It means a delay in simulation; when we use these data to predict potential floods, we need to add this time difference represented by the systematic error.

Comment 9.

Authors seem to know but are reluctant to provide more background information on using QPF or QPE on hydrologic simulations over Karst areas. Line 77-78 says that "only a few studies of rainfall forecasting based on WRF QPF and PERSIANN-CCS QPEs have been conducted in karst areas until now, and even if there are studies, the practical accuracy is generally poor." Are there any relevant studies? Reviewer believes there should be lots. What will be the differences between this one and other literature?

Response:

As mentioned earlier in our response to Comment 1, we agree that some relevant studies on WRF QPF and PERSIANN systems in karst basins have been conducted. However, among those studies using the WRF QPF model, the lead times of the models are short, usually no more than 24 hours. Such models with short lead times can forecast rainfall products with high accuracy. In contrast, the lead time of the WRF QPF model in this study can reach 96 hours. However, the model achieves quite high accuracy over such long lead times. The long response time achieved by coupling the WRF QPF model with the Karst-Liuxihe Model is very useful for flood forecasting. Many studies have been conducted on the PERSIANN system and its subsystem, PERSIANN-CDR (Climate Data Record) and obtained good results. However, only a few studies of PERSIANN-CCS (Cloud Classification System) QPEs in karst areas have been conducted.

These studies on WRF and PERSIANN models will be addressed in the introduction section of the revised manuscript, and the advantages of the WRF QPF and PERSIANN-CCS QPE models in this paper over other WRF models and PERSIANN systems will be discussed in the revised manuscript.

Comment 10.

Another concern is the investigation of parameter sensitivity. In this study, authors applied NSE to draw the conclusion about parameter sensitivity (Line 654-656 and Table 3) without perturbing each individual parameter. The current use of NSE only demonstrates how well the simulation can represent the observation. But, it has nothing to do with the sensitivity of parameter in the hydrologic model. More explanations of how authors evaluate the parameter sensitivity are needed.

Response:

The original sensitivity analysis of parameters was insufficient, and NSE is not adequate for evaluating parameter sensitivity. The parameter sensitivity analysis in the revised manuscript will be revised accordingly. More calculations and descriptions of parametric sensitivity analysis will be added to the revised version.

Minor issues:

Minor issues 1.

Abstract Line 20-27: This sentence runs for more than 8 lines without any clear structure. Please break into 2-3 smaller sentences for better readability.

Response:

The sentence will be divided into several shorter sentences for clarity.

Minor issues 2.

Line 66-68: this sentence is very cumbersome, The parts of "enable ... To be easily obtained" needs to be rewritten.

Response:

The sentence will be rewritten accordingly for clarity.

Minor issues 3.

Line 84-86: there are a few grammar errors in this sentence. Please re-write.

Response:

The sentence will be rewritten accordingly to correct grammar errors.

Minor issues 4.

Line 104-106: Ground gauge, of course, has no lead time as compared to the forecasting model. Not sure why authors emphasize on this known fact.

Response:

This fact was originally included for the convenience of non-professional readers. Since the reviewer thinks this is a known fact that need not be emphasized, it will be deleted from the revised manuscript.

Minor issues 5.

Line 108: grammar error. "People" cannot be "transferred". Do the authors mean "evacuate"?

Response:

Yes, the intended meaning was "evacuate". The word "transferred" will be replaced with "evacuate" accordingly in the revised version.

Minor issues 6.

Line 209: "The channel length of Liujiang river is about 1120 km and the area is about 5.8*104 km2". What area is about 5.8_104 km2 here? The area of water surface of the river?

Response:

The area of approximately $5.8*10^4$ km$^2$ is the basin area, not the area of water surface of the river. The sentence will be revised for clarity.

Minor issues 7.

Line 265&267: Why use the phrase "property data" here? What is the meaning of it?

Response:

The term "property data" here refers to DEM, soil type , and land-use type data. These three

kinds of data represent the basic properties of the underlying surface of the basin. In addition, these three data types are the most basic input property data for building a hydrological model.

Minor issues 8.
Line 350-351: The reviewer believes the authors are trying to say "the rainfall distributions of WRF QPF, the PERSIANN-CCS, and the observed Precipitation data appears to be quite similar to each other".

Response:

The reviewer is correct in that the intended meaning is "the rainfall distributions based on WRF QPF, PERSIANN-CCS and the observed precipitation data are similar to one another" The sentence will be revised accordingly.

Minor issues 9.
Line 385, and Line 399-404: In equation (1), the reviewer presumes the unit of Pi should be mm/cm/m, and the unit of Fi should be a mm2/cm2/m2. Thus, Pi times Fi should give us the volume of water here. Then Fi_Pi / N is still in volume. Later postprocessing procedure cannot be continued according to the instruction. It is important to give correct and detailed steps and evaluation of the applied postprocessing of two precipitation inputs.

Response:

The reviewer is correct. There was a calculation error in the original equation (1), the original equation is as follows:

$$\bar{P}_{\text{WRF/PERSIANN-CCS}} = \frac{\sum_{i=1}^{N} P_i F_i}{N}$$

After correction, it is as follows:

$$\bar{P}_{\text{WRF/PERSIANN-CCS}} = \frac{\sum_{i=1}^{N} P_i F_i}{A * N}$$

The unit of $Pi$ is mm, the unit of $Fi$ is mm$^2$, and $A$ is the basin area, mm$^2$. The units of these variables will be added in the revised version.

Minor issues 10.
Line 329-344 It seems like the authors did not evaluate precipitation product for the year of 2010. Is it because there was no flood event occurred in 2010? The following question would be why evaluate two precipitation product's performance only under several flood periods? Why not evaluate both precipitation products over as a long time period as possible?

Response:

The year 2010 was a dry year in the study area, and no major flooding events occurred during this year. In recent years, catastrophic floods have occurred frequently in the research area. In

the flood response, the flood peak is reached quickly and recedes quickly, which typically result in considerable damage. As shown in Fig. 10-16, the maximum peak flow was more than 25,000 $m^3$ $s^{-1}$; such flow caused a catastrophic flood event, and floodwater inundated many urban areas in the study area. The focus of this paper is to develop a method to accurately simulate and forecast flood events under flood periods to provide theoretical basis for flood control in karst areas. However, the suggestions of the reviewer are valuable, and we will study the WRF QPF and PERSIANN-CCS QPE with longer time periods in the future.

Minor issues 11.
Line 473: It might be better to use "improve" instead of "increase" in the section title.

Response:

The word "increase" in the title will be replaced with "improve" accordingly in the revised manuscript.

Minor issues 12.
Line 646 It might be better to say "the MPSA was modified and improved from the GLUE algorithm".

Response:

The text will be revised to "the MPSA was modified and improved from the GLUE algorithm" in the revised manuscript.

Minor issues 13.
Line 842: It is very difficult to tell the difference between the three types of rain gauges on Fig1a. Suggesting change high contrast color combination.
The abstract seems to be very long without a concise focus on the scope of work and summary of novelty and motivation of this study. It reads like an introduction instead of a concise abstract.
Unsupported claim on "Among these weather models, WRF QPF and PERSIANN-CCS QPEs may be better ways to acquire precipitation results effectively in karst basins."
Why WRF QPF and PERSIANN-CCS are selected for this study? References are needed.

Response:

The manuscript will be revised according to the above comments.
High contrast colour combinations will be used to distinguish the three types of rain gauges in Fig. 1a.
The abstract will be simplified in the revised version to focus on the summary and innovation of the research results.
The purpose of using the two weather models WRF QPF and PERSIANN-CCS QPEs in this study will be stated in the revised manuscript, and relevant references will be cited.

The studies cited in the responses to the reviewers' comments are as follows:
Chen, Y., Li, J., and Xu, H.: Improving flood forecasting capability of physically based distributed hydrological models by parameter optimization, Hydrol. Earth Syst. Sci., 20, 375–392,

https://doi.org/10.5194/hess-20-375-2016, 2016.

Chen, Y., Li, J., Wang, H., Qin, J., and Dong, L.: Large watershed flood forecasting with high-resolution distributed hydrological model, Hydrol. Earth Syst. Sci., 21, 735–749, https://doi.org/10.5194/hess-21-735-2017, 2017.

Li, J., Chen, Y., Wang, H., Qin, J., Li, J., and Chiao, S.: Extending flood forecasting lead time in a large watershed by coupling WRF QPF with a distributed hydrological model, Hydrol. Earth Syst. Sci., 21, 1279–1294, https://doi.org/10.5194/hess-21-1279-2017,2017.

Li, J., Yuan, D., Liu, J., Jiang, Y., Chen, Y., Hsu, K. L., and Sorooshian, S.: Predicting floods in a large karst river basin by coupling PERSIANN-CCS QPEs with a physically based distributed hydrological model, Hydrol. Earth Syst. Sci., 23, 1505–1532, https://doi.org/10.5194/hess-23-1505-2019, 2019.

---

## Referee Comment (RC2) · Anonymous Referee #2 · 24 Sep 2019

In this manuscript, the authors tried (1) to compare the performance of WRF QPF and PERSIANN-CCS QPEs and (2) to develop a new Karst-Liuxihe model. I agree with Reviewer 1 that it isn't convincing that the proposed new model can address the challenges of hydrological simulation in Karst areas. In addition, it isn't clear to how to add the karst mechanism into the Liuxihe model, and for example, regarding the karst water-bearing media simplification, it should be documented how to deal with this issue in the original model and how to improve it in the new model (described as equations or parameters). Also, I agree with Reviewer 1 that this manuscript isn't concise, especially too long abstract and introduction. In this manuscript, the authors tried to interpret too many issues so that the scientific contribution isn't clear. Consequently, it seems a

technique report. Therefore, I don't think that the current version is suitable to be published in this journal. Detailed comments 1. In Line 40, the authors states "to reflect the true conditions of rainfall-runoff", and what is "true conditions"? Does the original model describe not true conditions? 2. In Lines 77-78, some references for "a few studies" are required. 3. In Line 261, what's the meaning of "grid gauges"? 4. On the model simulating, it isn't clear how to obtain the information on karst fissure width and how to set the initial condition such as soil moisture. 5. In Lines 599-600, the authors introduced that total 30 floods were chose from 1982-2013 for verification, i.e. about 1 flood per year. The objective of this manuscript is flood forecasting, so it is better to choose more floods and evaluate the model according to the flood forecasting criteria. In addition, in Lines 600-60, I guess that the model used in (Li et al., 2019) isn't the new model and those results only implied the effectiveness of another model. Therefore, the sentences shouldn't be there. 6. In Figures 10-16, what is the unit of x-axis? h-1 or h?

---

## Author Comment (AC3) · 5 Oct 2019

Reply to the comments of Anonymous Referee #2-AC Anonymous Referee #2 In this manuscript, the authors tried (1) to compare the performance of WRF QPF and PERSIANN-CCS QPEs and (2) to develop a new Karst-Liuxihe model. I agree with Reviewer 1 that it isn't convincing that the proposed new model can address the challenges of hydrological simulation in Karst areas. In addition, it isn't clear to how to add the karst mechanism into the Liuxihe model, and for example, regarding the karst water-bearing media simplification, it should be documented how to deal with this issue in the original model and how to improve it in

the new model (described as equations or parameters). Also, I agree with Reviewer 1 that this manuscript isn't concise, especially too long abstract and introduction. In this manuscript, the authors tried to interpret too many issues so that the scientific contribution isn't clear. Consequently, it seems a technique report. Therefore, I don't think that the current version is suitable to be published in this journal. Detailed comments 1. In Line 40, the authors states "to reflect the true conditions of rainfall-runoff", and what is "true conditions"? Does the original model describe not true conditions? 2. In Lines 77-78, some references for "a few studies" are required. 3. In Line 261, what's the meaning of "grid gauges"? 4. On the model simulating, it isn't clear how to obtain the information on karst fissure width and how to set the initial condition such as soil moisture. 5. In Lines 599-600, the authors introduced that total 30 floods were chose from 1982-2013 for verification, i.e. about 1 flood per year. The objective of this manuscript is flood forecasting, so it is better to choose more floods and evaluate the model according to the flood forecasting criteria. In addition, in Lines 600-60, I guess that the model used in (Li et al., 2019) isn't the new model and those results only implied the effectiveness of another model. Therefore, the sentences shouldn't be there. 6. In Figures 10-16, what is the unit of x-axis? h-1 or h? We thank the referee very much for reviewing the manuscript. The following are our point-by-point responses to the reviewer's comments. Comment: Firstly, both reviewers thought it isn't convincing that the proposed new model can address the challenges of hydrological simulation in Karst areas. Response: This issue is mainly because the authors did not provide clear descriptions of the new model, especially in introduction. The abstract and introduction were quite cumbersome and failed to explain clearly the research motivation and innovation of this study. The abstract has been simplified to focus on summarizing the research and detailing the innovation, and the introduction has been restructured in the revised version. The most important innovation of the present research lies in the improvement and perfection of the Liuxihe model structure and function. A new hydrologic model, i.e., the Karst-Liuxihe model, is proposed, which has some enhancement modules added; this Karst-Liuxihe model has never appeared in the previous literature.

The improvement of Liuxihe model is extensive and includes the division of the model into the smallest structural units, termed karst hydrology response units (KHRUs); and new algorithms of rainfall-runoff, especially for the confluence of karst groundwater. This new Karst-Liuxihe model has some advantages in hydrological simulation in karst areas. The challenges of hydrological simulations in karst areas are mainly caused by the insufficient data supply. Distributed hydrological models usually have complex structures and numerous parameters, which require a large amount of hydrogeological data to build a model. However, it is difficult to obtain such data in karst areas due to the complex terrain.

Unlike other distributed hydrological models, the application of this Karst-Liuxihe model in the karst study area has certain data advantages due to its structural characteristics. There are only 3 layers in both the vertical subsurface and horizontal directions and the model structure is explicit, which makes modelling large data volumes less complex. Therefore, it is easy to build the Karst-Liuxihe model in karst area. The advantages of our model over other distributed hydrological models in karst hydrological simulation has been added to the revised introduction, and the related literature has also been cited and listed.

When the karst sub-basins are divided, other distributed models usually divide the whole karst area into a series of karst sub-basins according to DEM data. This strategy is appropriate for small karst basins but may not be suitable for such a large study area ($5.8 \times 10\hat{\,}4$ km2). In this study, the karst sub-basins were further divided into many smallest grid units known as karst hydrology response units (KHRUs) by the Karst-Liuxihe model. These KHRUs are small enough that spatial differences in the rainfall and terrain data of the underlying surface can be ignored, thus requiring less modelling data. The other challenge of distributed hydrological models in karst areas is the problem of model calculation efficiency. This study used an improved Particle Swarm Optimization method (Chen et al., 2016) to improve the calculation efficiency; however, a parametric uncertainty analysis had not been previous performed. A contribution

of this paper is that the parametric uncertainty is evaluated and specific and detailed parametric uncertainty calculations have been added to the revised version in section 6.2.

Comment: In addition, it isn't clear to how to add the karst mechanism into the Liuxihe model, and for example, regarding the karst water-bearing media simplification, it should be documented how to deal with this issue in the original model and how to improve it in the new model (described as equations or parameters). Response: The original Liuxihe model treats the entire underground layer as a whole, and the confluence calculation of the underground river is performed using a linear reservoir method. However, the karst groundwater system is obviously nonlinear. Therefore, the original Liuxihe Model cannot be used reliably in karst areas.

The Karst-Liuxihe model proposed in this study adapts to the complex hydrogeological characteristics of karst area by adding some karst mechanisms to the original Liuxihe model. Additional explanations and descriptions (will be described as equations and parameters) about adding karst mechanisms into the model have been added to the revised paper in section 4.2. In the revised section 4.2, methods for obtaining hydrogeological data of karst water-bearing media have been added; for instance, the permeability coefficient K was calculated by an experience function. In addition, in the revised paper, the parameters of the Karst-Liuxihe model and the value range of epikarst zone parameters are listed in Table 2. To demonstrate the advantages of the improved Karst-Liuxihe model, the flood simulation effects by the original Liuxihe model and the Karst-Liuxihe model have been compared in the revised version in section 6.1. Comment: Also, I agree with Reviewer 1 that this manuscript isn't concise, especially too long abstract and introduction. In this manuscript, the authors tried to interpret too many issues so that the scientific contribution isn't clear. Consequently, it seems a technique report. Therefore, I don't think that the current version is suitable to be published in this journal. Response: The abstract and introduction are quite cumbersome; thus, the scientific contribution was not clear. In the revised version, the abstract has

been simplified to focus on the contribution and the introduction has been restructured to clarify the motivation of this study and the novelty and methodology development of the model.

Detailed comments 1. In Line 40, the authors states "to reflect the true conditions of rainfall-runoff", and what is "true conditions"? Does the original model describe not true conditions? Response:

The true conditions of rainfall-runoff here refer to the true rainfall-runoff process in karst area. Maybe it is better to write "the true rainfall-runoff process". This sentence has been changed by "to reflect the true rainfall-runoff process". In general, rainfall-runoff conditions are more complicated in karst areas than non-karst areas. When dealing with surface runoff confluence, both the original Liuxihe model and the Karst-Liuxihe model are applicable. However, there is only 1 underground layer in the vertical direction in the original Liuxihe model, which treats the entire underground layer as a whole. This concept is based on the traditional lumped model when dealing with groundwater confluence. The confluence calculation of the underground river is a linear reservoir method. However, the karst groundwater system is nonlinear. Thus, the original Liuxihe model cannot describe the true underground runoff process. There are 3 underground layers in vertical structure in the Karst-Liuxihe modelïijŇincluding the soil layer, the rock stratum of the epikarst zone, and the underground river. Thus, the Karst-Liuxihe model is more suitable to depict the true rainfall-runoff process in karst areas. More details on the Karst-Liuxihe model structure improvements have been added in the revised paper.

2. In Lines 77-78, some references for "a few studies" are required. Response: Some necessary related studies have been added in the revised introduction. 3. In Line 261, what's the meaning of "grid gauges"? Response: The rainfall results calculated by these two weather models are based on the latitude and longitude of the location in the basin, and all the locations seem to form grids as shown in Figure 1a. Thus, the rainfall results are on the grids, and we called these locations grid gauges.

4. On the model simulating, it isn't clear how to obtain the information on karst fissure width and how to set the initial condition such as soil moisture. Response: For the model simulation, some of the settings for the initial values of the model were not clear. The hydrogeological information, such as the karst fissure width, was calculated by the drill-hole pumping test, and the permeability coefficient of the rock mass and the specific yield of the karst aquifer were calculated by an experience function according to the water inrush prediction of a coal mine in the study area. Some hydrogeological information, such as the distribution of karst conduits, fissures and cracks and the direction of underground rivers, were obtained through the tracing test in this study area. An additional explanation of acquired hydrogeological information, such as the karst fissure width, has been added to the revised paper section 4.2. For the model simulation, some initial conditions were determined by multiple model tests. For instance, the initial soil moisture is set to [0%,100%], with 0 indicating extremely dry soil and 100 indicating saturated soil water content. According to the effect of the flood simulation, the appropriate initial water content of soil was determined through model multiple calculations. In fact, based on our experience with models with multiple calculations in the study area, this initial soil moisture is usually 50%-80% during floods. Additional explanations of the initial conditions, such as soil moisture in the model, have been added to the revised version in section 5.1. For the model simulation, some initial conditions of the parameter optimization by the improved PSO algorithm have been determined in the revised paper in section 5.2.

5. In Lines 599-600, the authors introduced that total 30 floods were chose from 1982-2013 for verification, i.e. about 1 flood per year. The objective of this manuscript is flood forecasting, so it is better to choose more floods and evaluate the model according to the flood forecasting criteria. In addition, in Lines 600-60, I guess that the model used in (Li et al., 2019) isn't the new model and those results only implied the effectiveness of another model. Therefore, the sentences shouldn't be there. Response: The reviewer is correct. The 30 floods were simulated by the original Liuxihe model in Li et al. (2019). In the revised version in section 6.1, 20 karst flood events have been added and

simulated by the Karst-Liuxihe model and the Liuxihe model to test the performance of the improvement. The results are shown in Fig. 10 and Table 3 in the revised paper.

6. In Figures 10-16, what is the unit of x-axis? h-1 or h? Response: In Figs. 10-16, the unit of the x-axis should be h, which means hours. This problem also appears in Figs. 3-7. In the revised paper, these units in Figs. 3-7 and 10-16 have been revised accordingly.

Please also note the supplement to this comment:
https://www.hydrol-earth-syst-sci-discuss.net/hess-2019-285/hess-2019-285-AC3-supplement.pdf

―――――――――――――――――